# A Novel Group of Promiscuous Podophages Infecting Diverse Gammaproteobacteria from River Communities Exhibits Dynamic Intergenus Host Adaptation

Daniel Cazares,[a] Adrian Cazares,[b,c] Wendy Figueroa,[d] Gabriel Guarneros,[e] Robert A. Edwards,[f] Pablo Vinuesa[a]

[a]Centro de Ciencias Genómicas, Universidad Nacional Autónoma de México, Cuernavaca, Morelos, Mexico
[b]EMBL's European Bioinformatics Institute (EMBL-EBI), Wellcome Genome Campus, Cambridge, United Kingdom
[c]Wellcome Sanger Institute, Wellcome Genome Campus, Cambridge, United Kingdom
[d]Department of Biochemistry, University of Cambridge, Cambridge, United Kingdom
[e]Departamento de Genética y Biología Molecular, CINVESTAV IPN, Mexico City, Mexico
[f]FAME, Flinders University, Adelaide, SA, Australia

Adrian Cazares and Wendy Figueroa contributed equally to this work.

**ABSTRACT** Phages are generally described as species specific or even strain specific, implying an inherent limitation for some to be maintained and spread in diverse bacterial communities. Moreover, phage isolation and host range determination rarely consider the phage ecological context, likely biasing our notion on phage specificity. Here we isolated and characterized a novel group of six promiscuous phages, named Atoyac, existing in rivers and sewage by using a diverse collection of over 600 bacteria retrieved from the same environments as potential hosts. These podophages isolated from different regions in Mexico display a remarkably broad host range, infecting bacteria from six genera: *Aeromonas*, *Pseudomonas*, *Yersinia*, *Hafnia*, *Escherichia*, and *Serratia*. Atoyac phage genomes are ~42 kb long and highly similar to each other, but not to those currently available in genome and metagenome public databases. Detailed comparison of the phages' efficiency of plating (EOP) revealed variation among bacterial genera, implying a cost associated with infection of distant hosts, and between phages, despite their sequence similarity. We show, through experimental evolution in single or alternate hosts of different genera, that efficiency of plaque production is highly dynamic and tends toward optimization in hosts rendering low plaque formation. However, adaptation to distinct hosts differed between similar phages; whereas one phage optimized its EOP in all tested hosts, the other reduced plaque production in one host, suggesting that propagation in multiple bacteria may be key to maintain promiscuity in some viruses. Our study expands our knowledge of the virosphere and uncovers bacterium-phage interactions overlooked in natural systems.

**IMPORTANCE** In natural environments, phages coexist and interact with a broad variety of bacteria, posing a conundrum for narrow-host-range phage maintenance in diverse communities. This context is rarely considered in the study of host-phage interactions, typically focused on narrow-host-range viruses and their infectivity in target bacteria isolated from sources distinct to where the phages were retrieved from. By studying phage-host interactions in bacteria and viruses isolated from river microbial communities, we show that novel phages with promiscuous host range encompassing multiple bacterial genera can be found in the environment. Assessment of hundreds of interactions in diverse hosts revealed that similar phages exhibit different infection efficiency and adaptation patterns. Understanding host range is fundamental in our knowledge of bacterium-phage interactions and their impact on microbial communities. The dynamic nature of phage promiscuity revealed in our study has

Address correspondence to Daniel Cazares, cazares@comunidad.unam.mx, or Pablo Vinuesa, vinuesa@ccg.unam.mx.

implications in different aspects of phage research such as horizontal gene transfer or phage therapy.

**KEYWORDS** bacteriophages, environmental microbiology, experimental evolution, microbial ecology, phage-bacterium interactions

Bacteriophages, or phages for short, have been systematically described as species- or even strain-specific viruses (1–3). Since phage infection is initially conditioned to chance encounters with a receptive host cell, the premise of the narrow host range of the phages implies a severe limitation for them to be maintained and spread within environments featuring diverse microbial communities (4), which could promote the existence of phages with a wider range of potential hosts. Although several phages capable of infecting bacteria beyond the taxonomic rank of species or genus have been described, they are still few in number compared to the substantial volume of reports on genus- or species-specific phages (1–3).

Characterization of the host range is a paramount step toward assessing phage specificity and identifying specimens with a broad scope of host targets. Since the host range of a bacteriophage is described as the taxonomic diversity of hosts it can successfully infect (5), the strategies used to interpret a "successful infection" are key to characterize this trait. In order to infect, phages must attach to the host cell, transport their genomes into the cell, overcome various defense mechanisms, and hijack the host's molecular machinery to replicate, assemble, and release the new viral progeny (5, 6). Infection assays in liquid cultures, known as lysis curves, and spotting or plaque assays resulting in the formation of lysis spots or individual lytic plaques represent examples of commonly used methods to infer phage infection (2). As factors other than phages (e.g., pyocins) and mechanisms independent of phage production (e.g., lysis from without) can also lead to cell death by lysis, the isolation of individual lytic plaques, known as phage plaquing, is regarded as a reliable approximation to identify productive infections (i.e., an infection cycle yielding new phage progeny). Still, it is worth noting that plaque assays are susceptible to false-negative results in cases where phage infection do not lead to plaque formation (e.g., with reduced infection vigor or chronic infections) (2).

In spite of being labeled as infrequent, the isolation and study of phages with the ability to infect multiple species or genera can significantly contribute to advancing our understanding of the host range evolution and phage ecology. Recent metagenomics studies have provided valuable insights into the distribution of broad-host-range phages in different environments (5, 7, 8); however, the nature of this approach restricts further experimental characterization of these viruses which could target unculturable bacteria or strains that have not been isolated yet. On the other hand, some pioneering studies have focused on the targeted isolation of broad-host-range phages through enrichment procedures by either coculture (9) or consecutive monoculture (10) with potential hosts of different taxonomic origin. Such approaches led to the isolation of interesting unrelated specimens infecting bacteria from multiple species, including *Pseudomonas aeruginosa* and *Escherichia coli*.

Here we addressed the identification of broad-host-range phages by assembling a large collection of bacterial and phage isolates present in freshwater and wastewater samples. We hypothesized that using a taxonomically diverse panel of bacteria native to these environments as potential hosts would facilitate the isolation of phages infecting multiple species from such microbial communities. This approach led to the identification of a group of closely related podophages, named Atoyac (Nahuatl for river), capable of productively infecting bacteria from six different genera and three orders within the *Gammaproteobacteria* class. These phages, here referred to as "promiscuous" comprise a novel viral group with undetectable presence in available metagenomic data. The Atoyac phages display different efficiencies of plating among the genera they infect; however, such efficiency rapidly changes over consecutive propagation

in a single or multiple genomic backgrounds, indicative of dynamic host adaptation. The adaptation trajectories are phage dependent and lead to different outcomes despite the high similarity existing between the phages. Our findings contribute to expanding the virosphere and provide insights into short-term host range evolution and virus-host interactions in nature.

## RESULTS

**Isolation of promiscuous phages.** We searched for promiscuous phages, i.e., phages capable of infecting hosts of distinct taxonomic origin, by assembling a large and diverse collection of over 600 bacterial isolates and free phages existing in river and wastewater water samples from 12 different sites in Mexico (see Materials and Methods and Fig. S1 in the supplemental material). Serially diluted aliquots from most of the samples formed clearing zones and lytic plaques when spotted onto lawns of the panel of bacterial isolates retrieved from the water samples. The use of standard media such as TΦ or LB did not always allow the identification of phage plaques; however, utilization of diverse media and different dyes (see Materials and Methods) led to plaque visualization. A cross-infection screening among the susceptible isolates led to the identification of phages producing lytic plaques in multiple strains (see Materials and Methods and Fig. S1). Remarkably, six phages retrieved from five sampling sites were able to infect 54 bacterial isolates in the panel. Only these phages, which we named Atoyac, were selected for further analyses in this study. Susceptible bacteria included members of the genera *Aeromonas*, *Pseudomonas*, *Hafnia*, *Escherichia*, *Serratia*, and *Yersinia*, confirming the promiscuous nature of the Atoyac phage host range, which crosses multiple taxonomic ranks within the *Gammaproteobacteria* class (Fig. 1 and Fig. S2). The Atoyac phages were isolated from samples obtained from Central (Atoyac1, Atoyac10, Atoyac13, and Atoyac14), Southeast (Atoyac23), and Northwest (Atoyac15) regions in Mexico (see Data Set S1 in the supplemental material). Assessment of fecal contamination by either coliform count or presence of crAssphage (used as a marker to detect human fecal pollution in water [11]), revealed that the Atoyac phages were prevalent in contaminated sites, which represented most of our samples (Data Set S1). Two Atoyac phages (1 and 10) were isolated from sewage, and the other four were retrieved from rivers contaminated with sewage discharge. Contaminated samples were also the source of the highest phage titers determined in the plaquing assays.

**Efficiency of plating of the Atoyac phages.** The six Atoyac phages displayed similar host ranges. Differences in host range detected among the phages corresponded to the ability to infect some of the *Aeromonas*, *Hafnia*, and *Yersinia* isolates (Fig. 2A). Phage Atoyac15, for instance, was the only member of the group capable of infecting all the *Aeromonas* isolates in the panel but could not infect most of the *Yersinia* strains or some representatives of the *Hafnia* group.

We determined the efficiency of plating (EOP) (12) across the panel of susceptible strains to gain further insights into the infection capacity of the Atoyac phages and compared the EOP on each strain to the number of plaques formed on the reference strain *Aeromonas* sp. strain PIA_XB1_6 (isolated in this study and used as "reference" in estimating EOP) (Fig. 2). The results revealed complex EOP profiles among the susceptible isolates, although a clear pattern emerged. We found that clustering the bacterial isolates based on the phages' EOPs largely reflects their taxonomic relatedness at the genus level (Fig. 2A), suggesting that the efficiency of the Atoyac phages to produce PFU is influenced by the taxonomic origin of the host and that crossing the genus barrier imposes a cost. In general, Atoyac phages generated significantly more lytic plaques on *Aeromonas* isolates, followed by strains from the *Hafnia* and *Yersinia* genera, respectively (Fig. 2 and Data Set S2). No statistically significant differences in EOP per phage were detected between *Aeromonas* subclusters (Mann-Whitney U test and *P* value of 0.05), indicating that the major EOP differences occur between genera (Fig. 2B and Data Set S2). Some *Aeromonas* isolates featuring EOP values lower than those recorded in strains of other genera were also detected.

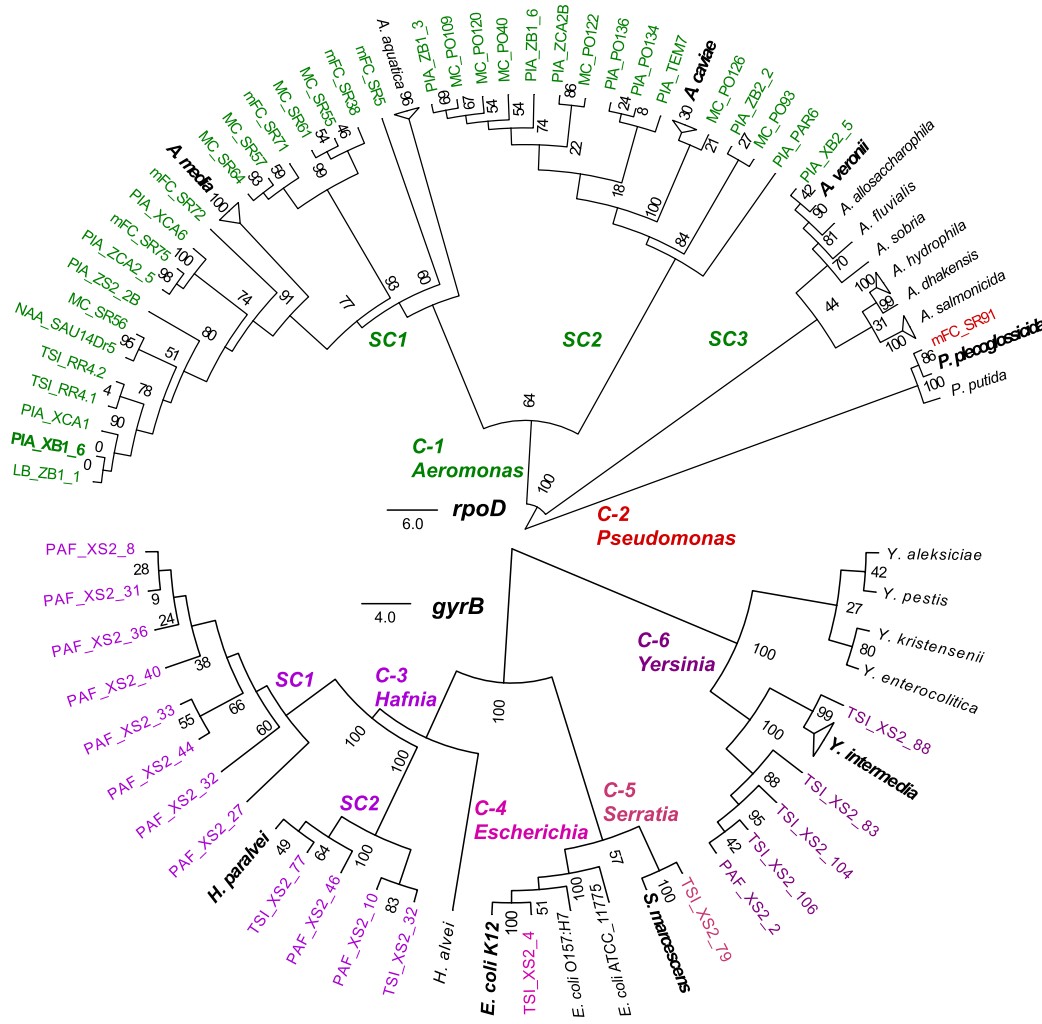

**FIG 1** Environmental bacterial isolates susceptible to infection by the promiscuous phages Atoyac. Neighbor-joining trees of the bacterial isolates infected by phages of the Atoyac group were reconstructed from the alignment of nucleotide sequences of the genes *rpoD* (top) or *gyrB* (bottom) under the J-C substitution model. Strains were considered susceptible only when Atoyac phages formed individual lytic plaques on them (see Materials and Methods). The 54 isolates from our collection are indicated in color labels corresponding to the bacterial genus they belong to. Reference strains of different bacterial species included in the trees are indicated in black labels, and the reference strains clustering with isolates from our collection are indicated in bold type. Clusters and subclusters of the different bacterial genera identified in the tree are indicated by the "C-" and "SC" prefixes, respectively. Branch support values from 100 replica bootstrap tests are shown in the trees. The scale bar represents the expected number of substitutions per site under this model.

The EOP values measured in the collection of susceptible strains ranged from −1,000 to 2.5 times the EOP on the reference strain *Aeromonas* sp. PIA_XB1_6 (original propagative strain, see Materials and Methods). For example, the stock of the Atoyac13 phage produced $6.72 \times 10^8$ PFU/ml on a bacterial lawn of *Yersinia* sp. strain TSI_XS2_88, $2.68 \times 10^{11}$ PFU/ml on *Aeromonas* sp. strain PIA_XCA1, and $1.04 \times 10^{11}$ PFU/ml on the reference strain *Aeromonas* sp. PIA_XB1_6 (Data Set S2).

**Virion morphology and genome comparison.** We characterized the Atoyac phages at the virion morphology and genome levels to investigate the similarities between them and to other phages. Electron microscopy analysis of the CsCl gradient-purified viral particles revealed a shared morphology typical of phages of the *Podoviridae* family, featuring an ~50-nm icosahedral head attached to a very short tail (Fig. 3 and Fig. S5).

The Atoyac phage genomes average 41.7 kb in size and 59% GC content. A

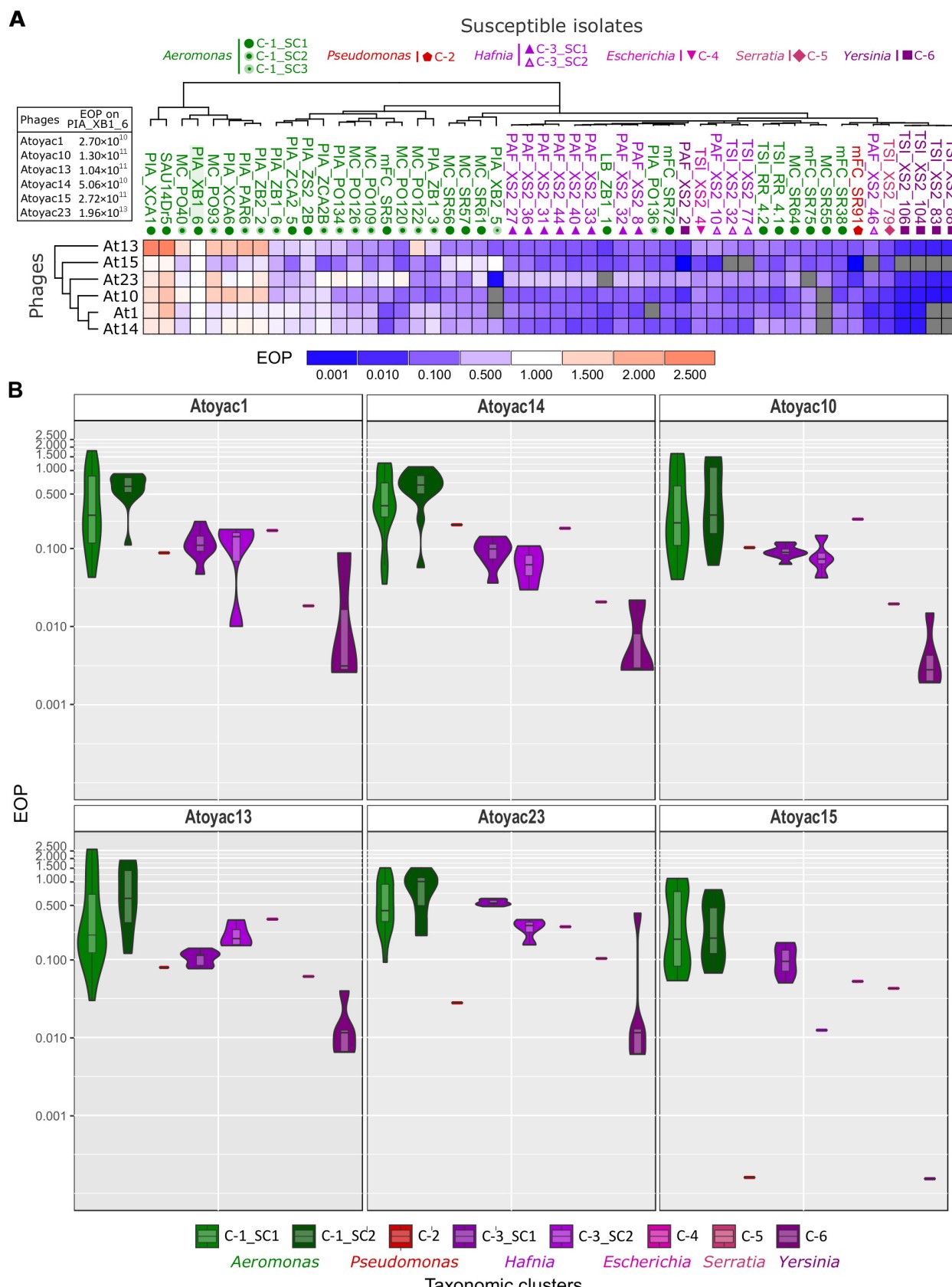

**FIG 2** Host range and efficiency of plating of the Atoyac phages. (A) The heatmap illustrates the efficiency of plating (EOP) recorded for six Atoyac (At) phages (rows) on a panel of 54 environmental strains isolated in this work (columns). Names of the bacterial isolates are color coded

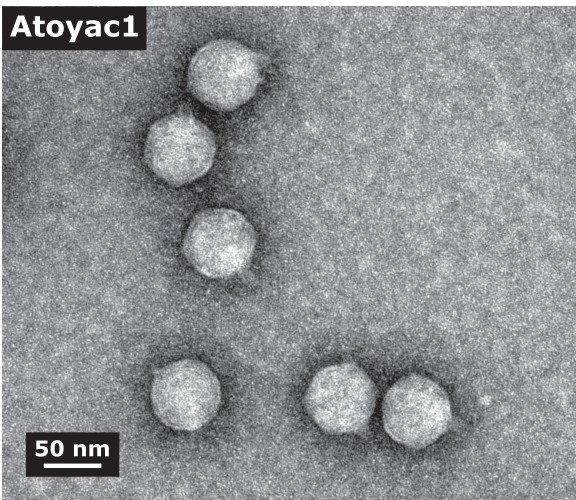

**FIG 3** Electron micrograph of virions of the Atoyac phage 1. The morphology of phage Atoyac1 is representative of those observed for all members of the Atoyac group. The CsCl-purified viral particles were negatively stained with 2% uranyl acetate and visualized at 150,000-fold magnification.

comparative analysis unveiled that they comprise a monophyletic group sharing from 85.9 to 98.4% overall nucleotide sequence identity (Fig. 4). We identified two sub-groups of phages (group 1, Atoyac phages 1, 14, and 10; group 2, Atoyac phages 13 and 23) that were more closely related to each other in the whole-genome tree and displayed very similar infection profiles. Phage Atoyac15 was the most divergent in the genome comparison and also exhibited one of the most dissimilar infection profiles. The major source of diversity among the Atoyac phage genomes corresponded to a large noncoding region featuring GC content lower than the global average and an adjacent cluster of open reading frames (ORFs) encoding mostly small (<100-amino-acids [aa]) hypothetical proteins of unknown function (Fig. 4 and Fig. S3). Marked sequence variation was also detected in contiguous ORFs putatively encoding a tail fiber and a glycosidase superfamily protein.

Of the 46 to 50 ORFs identified in the Atoyac genomes, 23 were assigned a function (46% to 50%) based on sequence homology and the presence of conserved domains. These ORFs are organized in two major functional modules encoding proteins related to (i) virion morphogenesis and release such as capsid, tail, and lysis proteins and (ii) transcription and replication, including RNA and DNA polymerases, a DNA primase, helicase, ligase, exonuclease, and endonuclease with a potential role in recombination (Fig. 4). Genes associated with a temperate lifestyle (e.g., repressor or integrase) were not identified in the genomes, implying that the Atoyac phages are virulent. This observation is consistent with our unsuccessful attempts to isolate lysogens of Atoyac phages in the *Aeromonas* and *Yersinia* backgrounds.

Homology searches revealed that the Atoyac phages constitute a novel group. Database searching revealed two groups of phages distantly related to the Atoyac clade. The most similar representatives from these groups correspond to the podophages 4vB_YenP_ISAO8 infecting *Yersinia enterocolitica* (hereafter referred to as

**FIG 2** Legend (Continued)
according to the taxonomic genus they belong to as indicated at the top of the figure. Symbols below the isolate names indicate the bacterial cluster or subcluster inferred through sequence alignment of a marker gene (see Fig. 1). The titer determined for each phage on the propagating strain PIA_XB1_6 (isolated in this study), indicated in the left inset, was used as the reference value to calculate the EOP. The scale at the bottom of the heatmap depicts the deviation from the EOP recorded in the reference isolate (EOP = 1.0). Gray cells correspond to interactions that did not generate detectable lytic plaques, thus indicating nonsusceptible strains. Both bacterial isolates and phages are hierarchically clustered based on the EOP values using the Euclidean distance method. (B) The EOP values recorded for each Atoyac phage grouped by the taxonomic cluster (genus) or subcluster of their hosts are plotted. Clusters are color coded as in panel A and Fig. 1. The EOP values were calculated from five independent biological replicates, and the averages are plotted. Values in the violin plots are in log scale.

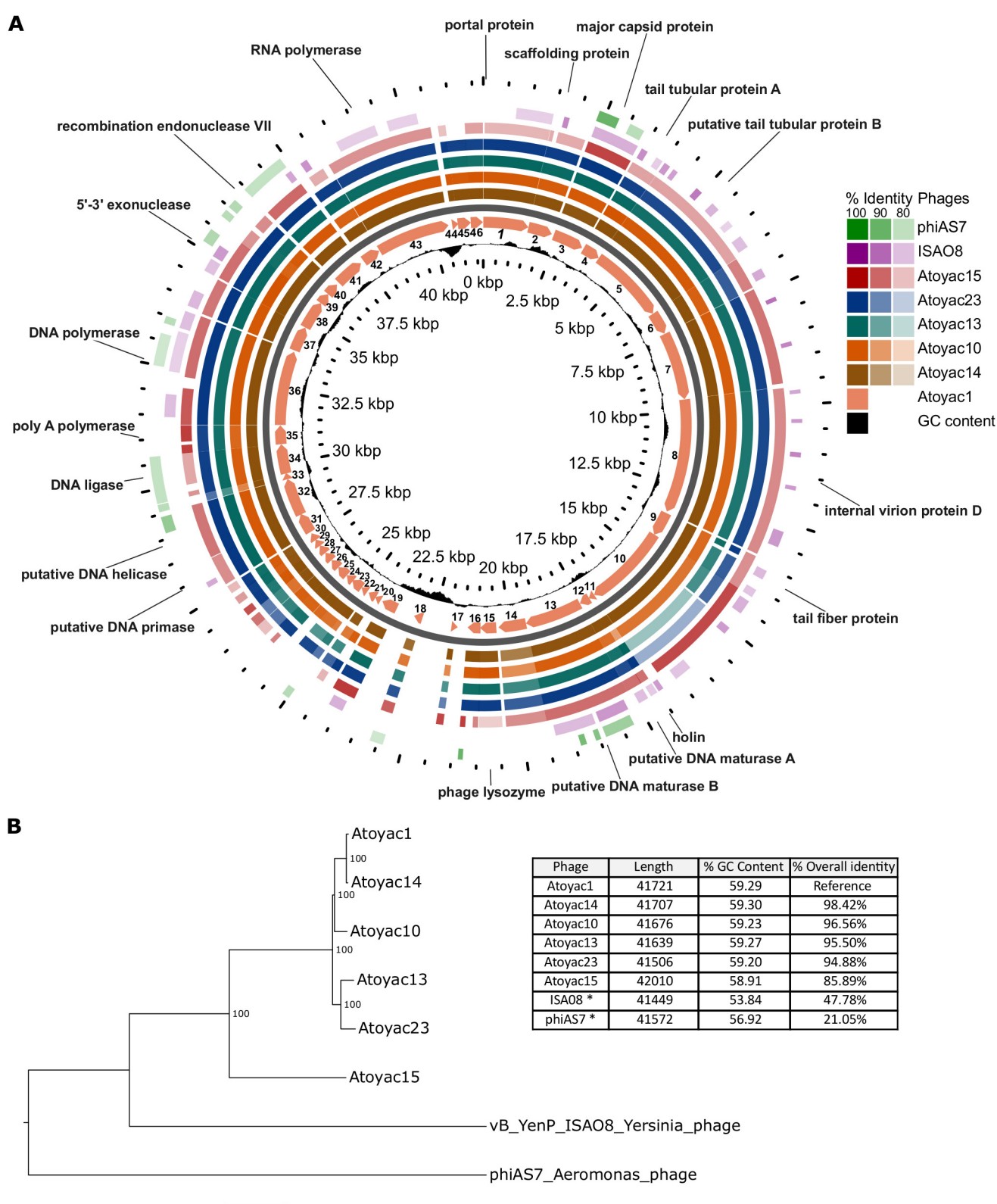

**FIG 4** Genomic comparative analysis and phylogenetics of the Atoyac phage group. (A) The genomes of six Atoyac phages and two distant homologues are represented as color rings and indicated to the right of the figure. The genome of phage Atoyac1, represented by a map in the innermost color ring, was used as a reference to compare the ORF sequences of the other phages at the nucleotide level. Arrows in the map correspond to ORFs pointing toward the direction of their transcription. Functions identified in the genome are indicated outside the rings pointing to the corresponding Atoyac1 ORFs. Coordinates of the reference genome and the distribution of its GC content regarding the average (59%) are indicated inside the corresponding ring. The

ISAO8, GenBank accession no. NC_028850.1) and phiAS7 infecting *Aeromonas salmoni-cida* (JN651747.1 [13]), which share 47.78 and 21.05% overall sequence identity with the genome of the phage Atoyac1, respectively (Fig. 4). We also compared the Atoyac phage genomes to that of the *E. coli* phage T7, a virulent archetype of the *Podoviridae* family. Despite the lack of extensive sequence similarity between the genomes at the nucleotide or protein level, the presence of a large nonhomologous and noncoding region was recognized in similar relative genome positions. A similar region has been identified in genomes of T7-like phages of different bacterial species and reported to contain multiple transcriptional promoters (14). In line with this observation, we identi-fied a series of putative promoters in both strands of the large noncoding region of the Atoyac genomes (Fig. 4 and Table S1).

**Search of Atoyac-like phages in metagenomics data.** Atoyac phages were identi-fied in different geographical regions in Mexico separated by ~3.9 to 2,525 km from each other, thus implying a broad distribution for this novel group of promiscuous viruses (Data Set S1). Since no other members of the group could be detected through homology searches in public genome databases, we decided to further investigate the distribution of the Atoyac phages by searching for similar sequences in available meta-genomics data.

We sought to recover sequencing reads homologous to the Atoyac1 genome by analyzing metagenome data sets deposited in the NCBI Sequence Read Archive (SRA). Surprisingly, our analysis of more than 65,000 data sets using a recently reported searching strategy (see Materials and Methods) (15) did not identify metagenomes with sequencing reads matching the Atoyac reference. This result contrasts with those obtained when genomes of other two unrelated phages, crAssphage and T4, were used as reference and control for the search. Thousands of reads covering up to 100 and 86.5% of the crAssphage and T4 genomes were recruited in multiple samples (Data Set S3), suggesting absence or low abundance of Atoyac-like phages in the ana-lyzed data sets. Alternatively, underexploration of environments inhabited by Atoyac-like phages (e.g., fresh waters) through metagenomics approaches could account for this outcome (8).

**Experimental evolution of efficiency of plating.** Since the Atoyac phages dis-played higher EOPs in isolates of the *Aeromonas* group than in strains of the other sus-ceptible genera, we took an experimental evolution approach to investigate the feasi-bility of reversing this trend in hosts with low plaque production efficiency. To diversify the set of hosts in the experiment, we chose *Yersinia* as representative of the *Enterobacteriaceae* family and *Pseudomonas* as the only non-*Enterobacteriaceae* genus identified as susceptible, besides *Aeromonas*. Isolates from the two selected genera recorded some of the lowest EOP values in the collection (Fig. 2A). Phages 1 and 23, dis-playing distinct EOPs in the bacterial isolates chosen for the experiment (*Pseudomonas* sp. strain mFC_SR91 and *Yersinia* sp. strain PAF_XS2_2; Fig. 2A), were selected as representa-tives of two subclusters identified within the Atoyac group (Fig. 4). Importantly, we allowed the phages to evolve by using ancestral bacteria at all times in the evolution experiments (see Materials and Methods).

Three individual plaques of Atoyac1 and Atoyac23, each representing a lineage, were serially propagated for 10 passages in a single host, *Pseudomonas* or *Yersinia* (see experimental design in Fig. S4, schemes A and B). To assess changes in plaque produc-tion over the course of the experiment, the EOP in *Pseudomonas* and *Yersinia* was cal-culated before the experiment and at propagation steps 5 and 10, using *Aeromonas* as

**FIG 4** Legend (Continued)
level of sequence identity detected in the genomes (>79%) respecting the Atoyac1 ORFs is color coded and indicated in the figure. The two outermost rings correspond to representatives of the phage groups displaying the highest sequence similarity to genomes of the Atoyac group. (B) The genomic tree of the Atoyac phages and two distant homologues (neighbor joining) was constructed from the alignment of the genomes at the nucleotide level. The bootstrap support value (100 replicates) of the branches is shown in each node. The overall sequence identity percentage (i.e., over the entire genome length) of Atoyac phages and their distant homologues (marked with an asterisk) regarding phage Atoyac1 (reference) is shown in the table, as well as the length and average GC content of each phage.

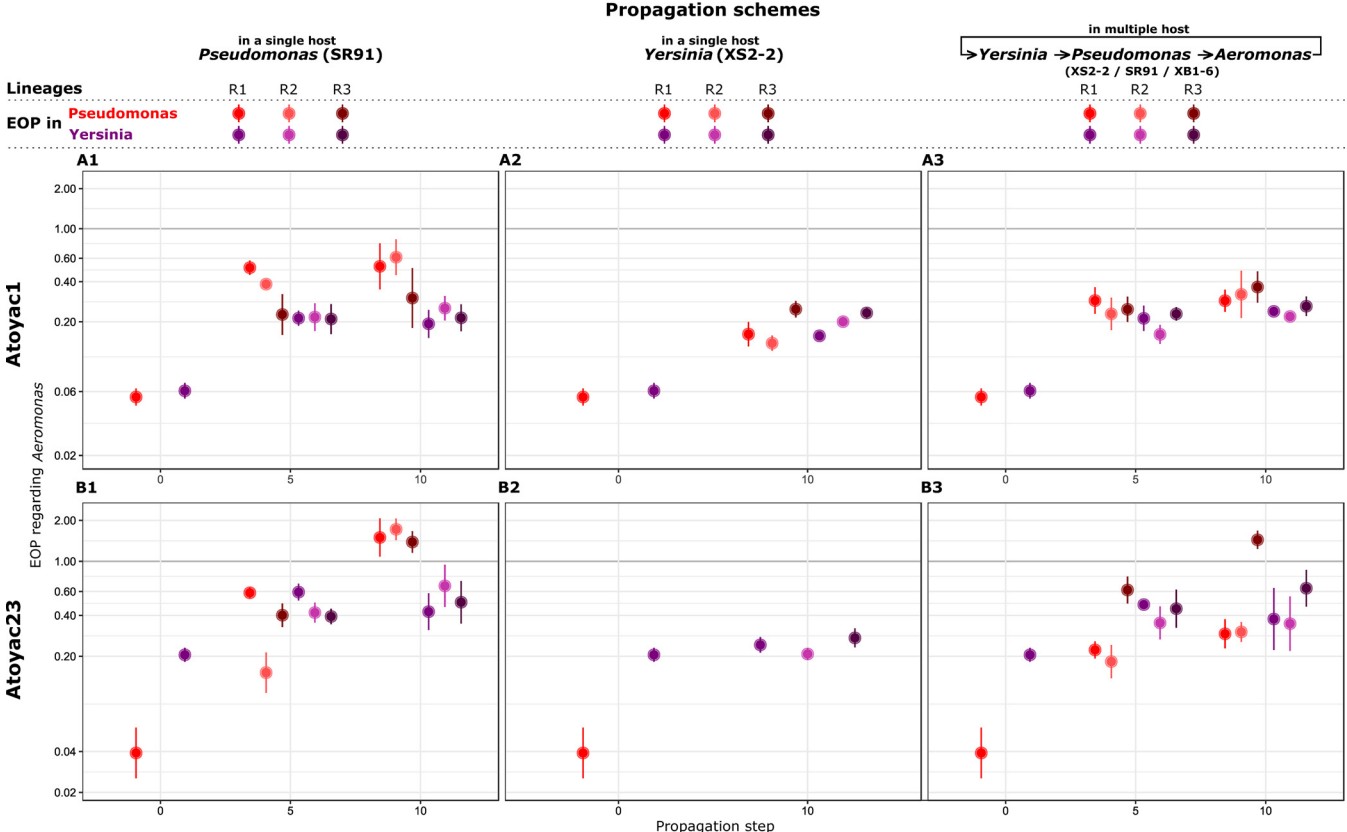

**FIG 5** Experimental evolution of the efficiency of plating of two Atoyac phages. The charts represent the trajectories of the dynamics of efficiency of plating recorded during the experimental evolution of phages Atoyac1 and Atoyac23 under two different propagation schemes (see Fig. S4 in the supplemental material). Note that in these experiments the bacteria used in every transfer corresponds to the ancestral host (see Materials and Methods); hence, only the virus was allowed to evolve. The top row (panels A1, A2, and A3) shows the outcomes recorded for phage Atoyac1. The bottom row (panels B1, B2, and B3) illustrates the results for phage Atoyac23. In the first column (A1 and B1), the charts show the EOPs calculated in *Pseudomonas* SR91 (red) or *Yersinia* XS2-2 (purple), using *Aeromonas* XB1-6 as the reference, when the phages were serially propagated in *Pseudomonas* for 0, 5, and 10 passages. Values plotted in propagation step 0 correspond to the EOPs of the ancestral phage stocks. The second column (B2 and A2) indicates the EOP trajectories recorded, when the phages were sequentially propagated in *Yersinia*. No EOP values for *Pseudomonas* are shown in panel B2 at propagation step 10 because we could no longer count plaques of the phage 23 on this host. Finally, the third column (A3 and B3) displays the results obtained when the phages were propagated alternating hosts between *Pseudomonas* SR91, *Yersinia* XS2-2, and *Aeromonas* XB1-6, at every transfer. R1, R2, and R3 represent independent lineages from the same ancestral phage stock. EOPs for every lineage were calculated from five technical replicates each. The filled circles show the averages, and the bars represent the standard deviations for the lineages. The y axes in all the charts are displayed in a log scale. All the data plotted in the charts are presented in Table S1 in the supplemental material.

the reference. As expected, the ancestral stock of both phages produced more plaques in the *Aeromonas* strain than in the isolates of the other two genera (Fig. 5). Nevertheless, this difference in efficiency of plating narrowed down in both *Pseudomonas* and *Yersinia* when the phages were consecutively replicated in the *Pseudomonas* isolate, suggestive of collateral evolution of expanded host range (Fig. 5, panels A1 and B1). A similar outcome was observed when Atoyac1 was serially propagated in the *Yersinia* isolate (Fig. 5, panel A2); however, Atoyac23 did not show a substantial change in EOP in the *Yersinia* strain and considerably reduced its virulence toward the *Pseudomonas* isolate (decrease in plaque number and size) under the same propagation conditions (Fig. 5, panel B2).

In parallel to the serial replication in a single strain, we also propagated the Atoyac phages cyclically alternating their host between *Yersinia*, *Pseudomonas*, and *Aeromonas* to explore whether this condition alters the dynamics in EOP resulting from continuous replication using a single host (Fig. S4, scheme C). We found that changes in EOP for both phages over the course of the experiment resembled those recorded when the phages are exclusively propagated in the *Pseudomonas* strain or when Atoyac1 is continuously replicated in *Yersinia*, i.e., a marked reduction in the EOP gap compared to the reference *Aeromonas* isolate (Fig. 5, panels A3 and B3). Interestingly, under this propagation scheme, Atoyac23 did not lose infectivity toward the *Pseudomonas* strain despite being replicated

in the *Yersinia* isolate in 4 of the 10 propagation steps, thus implying that host alternation may be key to maintain promiscuity.

## DISCUSSION

Phages are considered the most abundant and diverse biological entities on earth (16). Although numerous efforts exist to isolate and study new types of phages, a majority of them are targeted/biased toward isolating phages with a narrow host range (4), thus limiting our knowledge on phages with infection capabilities crossing between different taxonomic levels. These viruses are usually referred to as broad-host-range phages; however, we favor the use of "promiscuous" following the term applied to plasmids with mobilization and replication capabilities crossing multiple taxonomic levels (17). Moreover, the use of the term promiscuous allows distinction between viruses with hosts of different taxonomic origin and those infecting several strains of the same species, commonly referred to as broad range (5, 14). Promiscuous phages represent powerful models to further our understanding of phage-bacterium interactions and the evolution of host range; however, it is not yet clear how prevalent they can be (5).

Recent studies predicting phage-host relationships using various strategies have inferred interactions beyond the level of species or genus, suggesting that the abundance of promiscuous phages may be higher than currently postulated (4, 5, 7, 8). Previous attempts to isolate promiscuous phages primarily relied on enrichment methods using multiple bacteria of different species and mainly differ in either simultaneous or sequential addition of the potential hosts. These approaches led to the identification of phages infecting strains of two different genera or multiple species within a genus (9, 10). In these reports, either ATCC or other well-known type strains (e.g., *P. aeruginosa* PAO1 and *E. coli* K-12) were used as hosts (9, 10), which, although proven effective, limit the isolation to phages infecting strains that may not be representative of those existing in different environments. In contrast, our isolation strategy explored the use of a taxonomically diverse collection of bacterial isolates native to the sampled sites as potential hosts (see Fig. S1 in the supplemental material), under the rationale that isolating promiscuous phages is more likely when multiple bacterial strains used as indicators come from the same microbial communities they occur in.

Our isolation strategy allowed the identification of six phages, named Atoyac, infecting bacteria of the *Aeromonas*, *Hafnia*, *Yersinia*, *Escherichia*, *Serratia*, and *Pseudomonas* genera, which are part of the *Gammaproteobacteria* class. Interestingly, all the phages were retrieved from contaminated rivers. We speculate that in these environments there is a greater microbial diversity in high-density populations due to the presence of abundant resources available because of the constant human discharges, which would therefore favor the presence of phages capable of infecting different potential hosts. Nevertheless, we do not rule out that promiscuous phages could also be recovered from cleaner environments. We think our growth conditions could be applied to samples obtained from similar ecosystems. We encourage researchers to explore growth conditions that mimic the natural environment of the samples in order to retrieve bacterial isolates representative of the microbial diversity to maximize the possibility to find promiscuous phages. Following a similar approach regarding the use of native bacteria as potential hosts, other authors were able to isolate a phage from the Antarctic sea infecting strains of two genera from different proteobacteria classes (18).

Detection of phages infecting different species can be the result of a mix of multiple phages that have not been separated properly, leading to misidentification of a promiscuous host range (2). Host range assessment based on bactericidal effect (e.g., lysis spot) rather than productive infection can also lead to its overestimation (2). Here we addressed these issues through comprehensive evaluation of phage purity (Fig. S5) and determination of host range based on plaque production only. As previously recommended (2), our phage stock preparation included more than three rounds of plaque purification; additionally, the stocks were purified by cesium chloride (CsCl) density gradient prior to final host range determination (Fig. S1). Our experimental evolution

assays also involved 10 sequential passages in either single or multiple hosts of different genera (Fig. S4), conclusively demonstrating intergeneric propagation. In our experiments, we observed consistent evidence of phage stock purity (Fig. S5), including the presence of a well-defined band in the CsCl gradients, similar DNA restriction patterns and genome assembly sizes between phages, uniformity in plaque morphology, and homogeneity in virion morphology within and between samples.

Our results show that the six Atoyac phages have similar host ranges, consistent with their genomic similarity, and indicate that even the most divergent member of the group (Atoyac phage 15, displaying 86% overall sequence similarity to phage 1) is able to infect six bacterial genera. Interestingly, we found a good correlation between the clustering of the Atoyac phages based on the whole-genome comparison (Fig. 4) and that inferred from the EOP and host range profiles (Fig. 2A), suggesting that sequence variability largely accounts for the differential infection capability displayed by the phages. We identified a pair of putative structural genes encoding the tail fiber and a glycosidase superfamily protein, and two adjacent regions, one intergenic featuring putative promoters and one encoding a cluster of small ORFs of unknown function, that could account for the differences observed in the infection phenotypes given their sequence variation within the group. Mutations in virion structural proteins, particularly those that are components of the tail, have been reported as responsible for shifts in the host range of several phages, including in experimental evolution studies (19–21). Variable regions encoding accessory ORFs have also been documented in phage genomes (22); however, these have not been extensively characterized in terms of their functional impact on the phage biology.

Although the Atoyac phages infect six bacterial genera, the EOP analysis showed they produce more lytic plaques on *Aeromonas* isolates than in the other susceptible bacteria, suggesting that this phage group is better adapted to infect the *Aeromonas* genus (Fig. 2). The highly similar GC content between the Atoyac phages (59%) and *Aeromonas* (58 to 62% [23]) genomes and larger lytic plaques formed in the *Aeromonas* strains further support this notion (Fig. S2). An *Aeromonas* strain (PIA_XB1_6) was selected as the propagation host for the Atoyac phages because it was identified as highly sensitive to phage infection, along with other *Aeromonas*, in our initial screening to detect susceptible strains from the water samples. It is worth noting that propagation in PIA_XB1_6 prior to the final determination of host range could have also enhanced the EOP in this genus as a consequence of a potential domestication effect. Although we limited the number of passages to avoid this effect, it represents a potential artifact inherent to standard protocols to obtain pure phage stocks, a key requirement before assessing host range (2). Since the *Aeromonas* genus is known to be abundant in various aquatic environments like rivers (24), we hypothesize it serves as the primary host of the Atoyac phages, whereas isolates from the other susceptible genera, also present in rivers (25, 26), act as secondary hosts. We speculate that the presence of promiscuous phages in rivers may be related to the dynamics of this type of environment which is commonly influenced by strong seasonality (marked seasons of drought and heavy rain) that impact the composition of microbial communities diversifying it regularly (26, 27). Within this context, a promiscuous host range would be advantageous for the phage, as it would allow the use of alternative hosts to spread and remain in the community even if the primary host is not within its reach.

We observed statistically significant differences in the Atoyac phages' EOPs between *Aeromonas* and other susceptible genera but not between *Aeromonas* subclusters (see Data Set S2 in the supplemental material), suggesting that infection of distantly related bacteria involves a biological cost in terms of lytic plaque production. This notion is consistent with observations on other broad-host-range phages that exhibit substantial variation in EOPs when infecting distantly related hosts (9, 10). Variation in the efficiency of transfer or replication between distant hosts has also been reported for promiscuous plasmids (28), implying that the ability to spread among multiple taxonomic backgrounds does not come without a cost and that such trade-off is not exclusive of a certain type of mobile genetic element. However, despite the variation in the efficiency of plating, the

Atoyac phages did not show a considerable reduction in the EOPs in most of their hosts, even among the different genera (~85% of the isolates showed less than 100-fold EOP variations; Fig. 2). Bacteria often have intracellular defense strategies that allow them to limit infection by phages and other mobile genetic elements, the restriction-modification system and the CRISPR-cas system being some the most widespread antiphage defenses in bacterial genomes. These molecular mechanisms have been shown to dramatically impact the EOP by several orders of magnitude (2, 12). Therefore, it is possible that one or multiple putative ORFs of unknown function that were found during the genomic annotation and comparison in the Atoyac phages could encode mechanisms to counteract antiphage defense systems. A previous study by Jensen et al. (9) showed that broad-host-range phages that do not display significant decreases in their EOPs in different hosts (such as *Sphaerotilus natans* and *P. aeruginosa*) are insensitive to type I and II restriction enzymes. Similarly, the broad-host-range podophage T7 encodes the Orc protein that inhibits type I restriction enzymes as a mechanism to counterattack bacterial defense (29). Hence, we believe that by studying this type of phages, interesting proteins with biotechnology applications can be found, and it remains in our interest to study more in detail what the ORFs of unknown function in the Atoyac phages encode.

The Atoyac phages were isolated from river water samples obtained in different geographic regions across Mexico, suggesting a wide distribution of the group in similar environmental samples. Surprisingly, our search in metagenomics data did not identify sequencing reads mapping to the genome of the phage Atoyac1. While this outcome suggests the absence of the query genome in the explored data sets, it may also be indicative of the low abundance of Atoyac-like phages in the analyzed samples. The searching strategy used in this work involves a data set subsampling step to make the search through the SRA metagenomics data more efficient (15) with the cost of being susceptible to some false-negative results, particularly when the sequencing reads of certain taxa are in low abundance. Our study shows that culture-based approaches are still a good alternative to gain new insights on the virosphere diversity with the additional advantage of leading to the isolation of specimens that can be further investigated in the lab.

The comparative genome analysis identified the Atoyac phages as a novel viral group. Interestingly, the most similar genomes detected outside the group correspond to distantly related phages infecting either *Yersinia* or *Aeromonas* species, two of the bacterial groups identified as susceptible to infection by Atoyac phages in this work. Similarity matches to ORFs of phages with hosts of different genera (e.g., *Pseudomonas*, *Serratia*, *Escherichia*) can be detected by comparing sequences of conserved proteins encoded in the structural or replication modules of the Atoyac genomes; however, these matches feature low levels of sequence similarity, again indicating distant relationships (data not shown). Although we hypothesize *Aeromonas* as the primary host of the Atoyac phages, they display higher sequence similarity to the genome of the *Yersinia enterocolitica* phage ISAO8 (~48%) than to that of the *Aeromonas salmonicida* phage phiAS7 (~21%). It is unclear whether the phages ISAO8 and phiAS7 are specialized to infect their reported hosts or have a promiscuous host range as the phages of the Atoyac group. Further characterization of ISAO8 and phiAS7 could provide valuable insights into the evolution of the Atoyac phages' promiscuous host range in the future. Nevertheless, our study highlights the importance of comprehensively characterizing the host range of a phage by including taxonomically diverse hosts in the panel of tested strains. In the case presented here, the Atoyac phages could have been reported as *Aeromonas* "specific," given the large number of host strains identified within this genus, thus veiling the broader infection scope of the group.

Isolating promiscuous phages represents an unique opportunity to explore host range evolution among phylogenetically distant hosts; hence, we exploited this model to investigate the dynamics of infection of two Atoyac phages on nonoptimal hosts (*Pseudomonas* and *Yersinia*) using different propagation schemes. Our results show that successive propagation on either single or alternate hosts has a strong effect on

the EOP dynamics and its evolutionary trajectory, albeit the phage used is a critical factor in the outcome. In most cases, the serial propagation promoted EOP optimization in the inefficient hosts while maintaining the promiscuous host range, thus suggesting this is a well-conserved feature. The propagation of Atoyac23 exhibited the most contrasting results in the experiment. Continuous replication in *Pseudomonas* allowed the phage to reach an EOP and develop a plaque morphology, similar to those observed in *Aeromonas*, the initial most efficient host. In contrast, successive propagation on *Yersinia* did not have a substantial impact on the EOP in this host. Such dissimilar outcomes could be associated with the phage initial EOP in both hosts, i.e., Atoyac23 marked lower efficiency in *Pseudomonas* regarding *Yersinia*, and hence its margin for optimization. This notion is consistent with a recent experimental evolution study of a generalist cyanophage which acquired a larger number of evolutionary changes with a positive effect on infection efficiency when continuously propagated on a suboptimal host rather than an optimal host (21). Alternatively, evolutionary changes improving EOP in *Yersinia* might require more propagation steps to emerge and fix in the population.

Serial propagation of Atoyac1 in *Pseudomonas* improved its EOP in this host as well as in the *Yersinia* isolate, and vice versa. Phage Atoyac23, in contrast, displayed a substantial reduction in virulence toward *Pseudomonas* as a consequence of its continuous propagation in the *Yersinia* isolate. Replication in alternate hosts prevented the emergence of this effect, highlighting the importance of propagation in multiple hosts to maintain promiscuity in some phages. We speculate that similar propagation patterns could occur in certain environments featuring high microbial diversity, thus leading to preservation of phages with a promiscuous host range. Our results show that the adaptation outcome from continuous propagation on single or alternate hosts is largely dependent on the phage studied, even between highly similar phages, making the evolutionary trajectories difficult to predict.

## MATERIALS AND METHODS

**Sample collection and coliform count.** A dozen samples were collected in 1-liter sterile flasks from different rivers and sewage in central, northern, and southern Mexico. Physicochemical parameters such as temperature, pH, dissolved oxygen, etc., were measured *in situ* using a HANNA multiparametric HI9828 instrument (see Data Set S1 in the supplemental material). The recorded values were used as a guide for exploring our growth conditions in the laboratory. Estimation of water pollution in the different samples was performed by thermotolerant coliform counts on modified mTEC media, following the membrane filtration protocol previously described (30). Colonies producing a blue pigment (showing beta-glucuronidase activity) were counted, and the numbers of CFU per milliliter were calculated based on the dilution of the samples; these experiments were performed in triplicate. Sampling sites with coliform counts of $\geq 1 \times 10^3$ CFU/ml were classified as contaminated (Data Set S1).

**Bacterial isolation.** A diverse collection of bacterial isolates was assembled from the collected samples. Samples were plated onto different media: LB agar (Lennox), BHI (brain heart infusion) agar, PIA (*Pseudomonas* isolation agar), PAF (*Pseudomonas* agar F), PAP (*Pseudomonas* agar P), NAA (nutrient-poor medium) agar, mTEC (membrane thermotolerant *E. coli*) agar, mFC (membrane fecal coliform) agar, TSI (triple sugar iron) agar, and MC (MacConkey) agar, all manufactured by Difco and prepared following the manufacturer's instructions (31) and then incubated at 30°C for 24 h. After incubation, some colonies were picked to represent a variety of morphologies, sizes, and colors. The selected colonies were then purified by three consecutive passages on the same medium they were isolated from. Pure isolates were grown overnight in BHI broth at 30°C and stored at −80°C in 30% glycerol. The final collection was composed of around 600 isolates coming from the 12 different sampling sites.

**Phage isolation.** Bacterial lawns of the isolates in our collection were prepared by mixing 80 to 250 µl of overnight cultures with 3.5 ml of top agar (1% peptone, 0.5% NaCl, 0.7% Bacto agar, 10 mM MgSO₄), and plating the mixture on TΦ (1% peptone, 0.5% NaCl, 1.1% Bacto agar), LB, BHI, mFC, TSI, MC, and MC without salts media (31) to find the best conditions for identification of phage plaques (the media used for optimal growth and visualization of plaques in the strains identified as susceptible to the Atoyac phages are listed in Data Set S2). Aliquots of the water samples used to isolate bacteria were centrifuged, filtered (MF-Millipore membrane filter, 0.22-µm pore size), serially diluted, and then spotted on the bacterial lawns to identify strains allowing the formation of isolated lytic plaques, indicative of susceptibility to phage infection. Plates were incubated overnight at 30°C. To aid the visualization of the lytic plaques, rosolic acid (0.01%), phenol red (0.0024%), or bromophenol blue (0.0024%) were used in some of the media (TΦ, LB, BHI, TSI, MC, and mFC). The concentrations of dye used are based on those reported for Difco's media such as mFC and TSI (31).

Cross-infection assays were then used to identify candidate promiscuous phages (i.e., phages that

could infect a wide diversity of isolates). Briefly, individual plaques were picked with a sterile toothpick and streaked onto bacterial lawns of the strains previously identified as phage susceptible. To assess whether infection of a given strain affected infectivity in other hosts, the cross-infection assays were performed in two different ways: simultaneously, i.e., one selected plaque was transferred to the other bacterial isolates in parallel, and sequentially, i.e., a plaque produced in one tested bacterial isolate was transferred to the next potential host (see Fig. S1 in the supplemental material). Phages infecting multiple isolates were considered candidates to promiscuous phages, but only those with the broadest host range were selected for further characterization. These phages were then purified by four sequential passages in the strain PIA_XB1_6 (identified taxonomically as *Aeromonas* sp.), in which the phages exhibited some of the highest titers, largest plaque size, and a short plaque generation period (~5 h) during the first screening to identify phage-susceptible strains; hence, it was considered the optimal propagation host. Additional purification passages were avoided to reduce a possible domestication effect of phages in this host. High-titer stocks were generated from the last purification pass as described previously (22).

**Taxonomic identification of bacterial isolates to genus level.** Bacterial isolates that were susceptible to infection by phages were grown on LB overnight at 30°C. Genomic DNA was extracted from the cultures by following the protocol previously reported by Chen and Kuo (32). The extracted DNA was used as the template for PCR amplifying the genes *gyrB* and *rpoD*. We favored the use of the housekeeping genes *rpoD* and *gyrB* over the 16S rRNA gene as markers because the former have a higher evolutionary rate and have shown better results to resolve complex groups such as *Pseudomonas*, enterobacteria, and *Aeromonas* to the intrageneric level (33–35).

Oligonucleotides to amplify the *gyrB* (forward, 5′-GGCGGTAARTTYGAYGATAAC-3′; reverse, 5′-TAGCCTGGTTCTTACGGTT-3′′′; reported here) were used in those isolates that displayed phenotypic traits similar to those of the *Enterobacteriaceae* family (e.g., growth in MC plates). Primers flanking *rpoD* (forward, 5′-ATYGAAATCGCCAARCG-3′; reverse, 5′-CGGTTGATKTCCTTGA-3′ [36]) were used with the rest of the bacterial isolates. PCR assays were performed using Thermo Scientific *Taq* DNA polymerase with 0.2 $\mu$M each primer and the following cycling conditions: 95°C for 3 min; 30 cycles with 1 cycle consisting of 95°C for 30 s, 52°C for 1 min for *gyrB* and 56°C for 1 min for *rpoD*, 72°C for 1 min; and 72°C for 10 min. PCR products were purified using MultiScreen PCR$\mu$96 and sequenced by Sanger sequencing in Macrogen, Inc. The bacterial genera were determined by performing a BLASTn search of the obtained sequences against the NCBI database in order to identify the genera of the closest matches. The homologous sequences (using only type strains as the reference) were aligned using Clustal omega and used to generate neighbor-joining trees (37).

**Purification of promiscuous phages.** Phage stocks were treated with DNase I and RNase (1 $\mu$g/ml) and incubated for 30 min at 37°C. Phage particles were then precipitated with 16% (wt/vol) polyethylene glycol (PEG) 8000 and 1.4 M NaCl overnight at 4°C and centrifuged at 8,000 relative centrifugal force (RCF) for 20 min. The supernatants were discarded, and the pellet was resuspended in 1 ml of SM buffer (50 mM Tris HCl [pH 8], 10 mM MgSO$_4$, 100 mM NaCl, and 0.01% gelatin). In order to remove the PEG from the stocks, an equal volume of chloroform was added to the tubes and centrifuged at 8,000 RCF for 5 min. The aqueous phase was recovered and purified by ultracentrifugation in a discontinuous CsCl gradient as previously described (38). Finally, the pure phage stocks were dialyzed to remove the salts and stored at 4°C.

**Determination of phage host range and efficiency of plating.** Cross-infection assays (see "Phage isolation" section above) were used to characterize the host range of the promiscuous phages based on their ability to produce isolated plaques in the different bacterial isolates. Phages capable of infecting different taxonomic genera were classified as promiscuous and named Atoyac, which means river in Náhuatl (an indigenous language of Mexico). To determine the EOP of the Atoyac phages, serial dilutions of the CsCl-purified stocks were spotted onto the susceptible isolates of the bacterial collection. The EOP was calculated as the average of the titer of the phage in a given strain divided by the titer in the isolate that was used to initially propagate the phage (PIA_XB1_6). Each plaque assay was performed with five replicates.

**Statistical analysis.** In order to determine whether the data of efficiency of plating was normally distributed, histograms, quantile-quantile (QQ) plots, and Shapiro-Wilk tests were used taking as input the EOP values of each phage on the different bacterial genera. As the data were not normally distributed due to the nature of the samples, the nonparametric Mann-Whitney U test was used and *P* values were corrected for multiple comparisons using the Bonferroni correction. Corrected *P* values lower than 0.05 were considered statistically significant (Data Set S2). For the experimental evolution data, one-way analysis of variance (ANOVA) tests were used to compare the difference of EOP for each condition among the different time points. *P* values lower than 0.05 were considered statistically significant (Data Set S4).

**Virion morphology.** The virion morphology was determined by negative-contrast stain transmission electron microscopy as previously described by Cazares et al. (22). Briefly, aliquots of the pure phage stocks were deposited on a carbon-coated copper grid and stained with 2% uranyl acetate. Grids were examined under a JEM-2000 transmission electron microscope at 80 kV.

**Phage DNA extraction and genome sequencing.** DNA from the six Atoyac phages was extracted from 500 $\mu$l of CsCl-purified phage stocks following the protocol previously described (38), excluding the step with proteinase K. Purity and concentration values were measured by spectrometry (Nanodrop 2000). Aliquots of 500 ng of DNA were used to determine the size of the genome by restriction digestion. Finally, the DNA for the different phages was sequenced by Illumina (paired-end) technology in the Unidad Universitaria de Secuenciación Masiva y Bioinformática (UUSMB)-Universidad Nacional Autónoma de México (UNAM).

**Phage genome analysis.** The reads were adapter trimmed and quality filtered using Trimmomatic v.0.36 (39). The filtered reads were then assembled using Velvet v.1.2.10 (40) and mapped against the *de novo* assemblies using BWA v.0.7.17 (41). The files with the mapped reads were then used to correct the assemblies using Pilon v.1.22 (42). The genomes were preliminary annotated with Prokka v.1.13 (43) and reannotated manually based on protein homology searches and conserved domains using InterProScan 5.30-69.0, PSI-BLAST, and CD Search (44–46).

The genomes were used for comparative genomics analysis of the Atoyac phages and their homologous, 4vB_YenP_ISAO8 (GenBank accession no. NC_028850.1) and phiAS7 (JN651747.1) phages. The coding sequence (CDS) regions of Atoyac1 phage were used as reference to compare with the other phages using the GView Server (https://server.gview.ca/ [47]) and generate a comparative circular map (BLAST Atlas). Similarly, Atoyac phages were aligned using Mauve v.2.4.0 with the default parameters to create a comparative linear map and closely visualize the main variable regions of the genomes (48). Finally, the phages' genomes were globally aligned with Clustal omega and a BioNeighbor-Joining tree was constructed based on the alignment (37).

**Promoter prediction.** The *in silico* promoter search for the intergenic region was performed using the phage Atoyac1 sequence (from nucleotide 21377 to 23783), as the reference of the group, with the promoter prediction tools (for both strands): Neural Network Promoter Prediction (NNPP), PromoterHunter, and BPROM through their web servers (www.fruitfly.org/seq_tools/promoter.html, www.phisite.org/main/index.php?nav=tools&nav_sel=hunter, and www.softberry.com/berry.phtml?topic=bprom&group=programs&subgroup=gfindb [49–51]).

**Search of Atoyac genomes in metagenomics data.** The genome of phage Atoyac1 was used as a reference to search for homologues in metagenomics data deposited in the NCBI Sequence Read Archive (SRA). Whole shotgun metagenomes available from SRA and previously curated with PARTIE (52) were mapped to the reference genome using bowtie2 (53) with default parameters through the Search SRA Gateway (https://www.searchsra.org/) (15, 54, 55). Only alignments of reads > 50 bp were considered. The number of mapped reads and coverage of the alignments were determined from the sorted and indexed BAM files using Samtools v.1.7 (15, 48, 56).

**Experimental evolution of efficiency of plating.** Our experimental evolution strategy consisted of the sequential propagation of the phages Atoyac1 and Atoyac23 by plaque assays under two different schemes (Fig. S4). In the first scheme, phages were propagated in a single host, using either the *Pseudomonas* strain SR91 or *Yersinia* XS2-2, in which both phages recorded low EOP regarding the *Aeromonas* strain XB1-6 used as the reference to calculate the efficiencies. In the second scheme, phages were propagated in multiple hosts, cyclically alternating the use of the strains XS2-2, SR91, and XB1-6. Atoyac1 and -23 were selected as the most divergent representatives of two subgroups identified within the Atoyac group (excluding Atoyac15 which represents the outlier [Fig. 4]) and because they exhibited distinct EOP profiles in the bacterial hosts chosen for the experiments. In both evolutionary schemes, the phages were propagated for 10 passages starting from three isolated plaques (lineages) obtained by plating the ancestral phage stock on bacterial lawns of the selected hosts. In every transfer, overnight cultures of the ancestral bacteria, grown from frozen glycerol stocks, were used to prepare the lawns utilized in the plaque assays; therefore, only the virus was allowed to evolve in the experiments. EOPs of the phages were calculated at propagation steps 0, 5, and 10, using high-titer phage stocks generated in these stages and plating them onto lawns of the ancestral *Pseudomonas*, *Yersinia*, and *Aeromonas* strains. The titer recorded in *Aeromonas* in the corresponding propagation step was used as reference to determine the EOP, which was calculated from five technical replicates (see Fig. S4 for experimental design).

**Data availability.** The Illumina-sequenced genomes and annotations of the Atoyac1, Atoyac10, Atoyac13, Atoyac14, Atoyac15, and Atoyac23 phages were deposited in GenBank under accession numbers MT682386 to MT682391. The six Atoyac phages and three of their hosts (*Aeromonas* PIA_XB1_6, *Yersinia* PAF_XS2_2, and *Pseudomonas* mFC_SR91) were deposited in the Félix d'Hérelle Reference Center for Bacterial Viruses.

## SUPPLEMENTAL MATERIAL

Supplemental material is available online only.
**DATA SET S1,** XLSX file, 0.02 MB.
**DATA SET S2,** XLSX file, 0.1 MB.
**DATA SET S3,** XLSX file, 0.1 MB.
**DATA SET S4,** XLSX file, 0.04 MB.
**FIG S1**, TIF file, 0.3 MB.
**FIG S2**, TIF file, 0.8 MB.
**FIG S3**, TIF file, 0.5 MB.
**FIG S4**, TIF file, 1 MB.
**FIG S5**, TIF file, 2.5 MB.
**TABLE S1**, XLSX file, 0.01 MB.

## ACKNOWLEDGMENTS

D.C. gratefully acknowledges the Programa de Doctorado en Ciencias Biomédicas, UNAM, and the Ph.D. scholarship 586079 from Consejo Nacional de Ciencia y Tecnología (CONACYT, México). A.C. has been supported by the EMBL-EBI/Wellcome Trust Sanger Institute Join Post-Doctoral Fellowship Program (ESPOD). P.V. acknowledges funding from DGAPA-PAPIIT/UNAM IN206318 and CONACYT 23_A1-S-11242. G.G. acknowledges funding from CONACYT CB 255255. W.F. acknowledges funding from Cambridge Trust-CONACYT 706017.

We thank María de Lourdes Rojas-Morales from Microscopía Electrónica, CINVESTAV, for technical assistance in our electron microscopy studies and Javier Rivera Campos from CCG, UNAM, for technical support with lab experiments.

We declare that we have no competing interests.

D.C., A.C., W.F., and P.V. conceptualized the study. D.C., A.C., and W.F. drafted the manuscript with input from the other authors. D.C. isolated the phages and carried out the characterization experiments and data analysis and visualization. A.C. contributed to the genome and metagenome data analysis, virion purification, and characterization by electron microscopy. W.F. performed the statistical analyses and contributed to virion purification, phage DNA extraction, and characterization by electron microscopy and EOP. G.G. contributed by supervising the work. R.A.E. contributed to the metagenomics data analysis and revision of the manuscript. P.V. was responsible for funding acquisition and contributed to the phylogenomic analysis.

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
