## [Reviewer comments · mSystems]

A novel group of promiscuous podophages infecting diverse Gammaproteobacteria from river communities exhibits dynamic inter-genera host adaptation

Daniel Cazares, Adrian Cazares, Wendy Figueroa, Gabriel Guarneros, Rob Edwards, and Pablo Vinuesa

Corresponding Author(s): Daniel Cazares, Center for Genomic Sciences, UNAM

Review Timeline:

Submission Date:	August 6, 2020
Editorial Decision:	September 14, 2020
Revision Received:	December 15, 2020
Editorial Decision:	December 30, 2020
Revision Received:	January 12, 2021
Accepted:	January 13, 2021

Editor: Seth Bordenstein

Reviewer(s): Disclosure of reviewer identity is with reference to reviewer comments included in decision letter(s). The following individuals involved in review of your submission have agreed to reveal their identity: Antonet Svircev (Reviewer #1)

Transaction Report:

DOI: <https://doi.org/10.1128/mSystems.00773-20>

September 14, 2020

Dr. Daniel Cazares
Center for Genomic Sciences, UNAM
Genome Engineering
Cuernavaca, Morelos 62210
Mexico

Re: mSystems00773-20 (Dynamics of infection in a novel group of promiscuous phages and hosts of multiple bacterial genera retrieved from river communities)

Dear Dr. Daniel Cazares:

Below you will find the comments of two expert reviewers who provided several suggestions and feedback on your manuscript. In addition to their comments, mSystems prefers to have a limited number of supplementary figures. Please consolidate them, delete any if possible, and/or move several of the supplementary figures to the main text in your revision; provide an update in the response letter.

To submit your modified manuscript, log onto the eJP submission site at <https://msystems.msubmit.net/cgi-bin/main.plex>. If you cannot remember your password, click the "Can't remember your password?" link and follow the instructions on the screen. Go to Author Tasks and click the appropriate manuscript title to begin the resubmission process. The information that you entered when you first submitted the paper will be displayed. Please update the information as necessary. Provide (1) point-by-point responses to the issues raised by the reviewers as file type "Response to Reviewers," not in your cover letter, and (2) a PDF file that indicates the changes from the original submission (by highlighting or underlining the changes) as file type "Marked Up Manuscript - For Review Only."

Due to the SARS-CoV-2 pandemic, our typical 60 day deadline for revisions will not be applied. I hope that you will be able to submit a revised manuscript soon, but want to reassure you that the journal will be flexible in terms of timing, particularly if experimental revisions are needed. When you are ready to resubmit, please know that our staff and Editors are working remotely and handling submissions without delay. If you do not wish to modify the manuscript and prefer to submit it to another journal, please notify me of your decision immediately so that the manuscript may be formally withdrawn from consideration by mSystems.

Sincerely,

Seth Bordenstein

Editor, mSystems

Journals Department
Reviewer comments:

Reviewer #1 (Comments for the Author):

The manuscript describes the isolation, characterization, host range and genomics of Podoviridae bacteriophages, named by authors Atoyac, from sewage and rivers in Mexico.

GENERAL COMMENTS

- 1) The title and abstract fail to describe the major discoveries of the study. Both are too broad and do not reflect the obtained results.
- 2) Authors are encouraged to use terminology that is accepted by majority of bacteriophage researchers. I highly encourage the authors to consult all of Steve Abedon's publications on bacteriophages, paying particular attention to his work on bacteriophage host range and the limitations of plaque assays in determining phage host range. Please do not over use the term "promiscuous" when in fact you are refereeing to broad host range bacteriophages.
- 3) The manuscript contains non-standard terminology; terminology should be explained to the reader or referenced.
- 4) If I understand this correctly, out of all the bacteriophages isolated (54?) only 6 were sent for full genomic sequencing? If so, are the genomes deposited in GenBank? Materials section does not provide this information. Biological replicates should be mentioned in the M&M, there was a brief mention in a figure description.
- 5) Bacterial genes rpoD and gyrB were used to bring the bacterial isolates down to genus level. In my experience and based on literature, 2 genes are not enough.
- 6) The Materials section is missing important information. While it should be succinct, nonetheless, the reader should be able to repeat and understand what was done and how it was done.
L489 What do all the media short terms refer to? What percentages of the media were used? Who is the manufacturer?
L491 "Millipore membrane"????L495 is this from some paper? Reference?
L496 "cyclic and simultaneous cross infection" definition?reference?
L531 "Cross infection assay, see above"?
L524 "Pelleting at 8,000 G" Meaning?
- 7) Samples were collected from sewage/river samples (L469-476). River water samples with high coliform counts used. Would not sewage have all the bacteriophages you need? I was not clear as to why the river waters were collected. In the rivers were you looking for human origin coliforms?
Was this study
- 8) This manuscript has too many supplements. Supplements should be used for complex/detailed data that confirms main results - otherwise it should be simply 'data not shown'.

9) In the Results sections authors should only describe the results that were described in the figures/data tables etc. The focus should be on the material in the main manuscript, not the supplements. L115 "phages isolated from river and sewage.. (Supplement Fig. 3)" Is the reader really supposed to go to this? What is missing here in the main body of the results, is the number of phages isolated from river and numbers isolated from river which were further characterized in this work. L189-191 this should be in the discussion.

SPECIFIC COMMENTS

L26 "bacteria isolated from the same samples", not clear what is meant.

L32-33 and L94 "plating efficiency" or efficiency of plating (EOP)?

L60-63 This is not a correct statement. Literature contains a significant number of publications that describe the ability of the Enterobacteriaceae phages to infect different genera.

L104 "infectivity assays" are actually spot tests.

L105-106 What does this mean.

L107-109 Meaning of this sentence is not clear. This study simply relied on spot tests and toothpick streaks.

L109 What is "cross-infection screening"?

L111 The study actually focuses on 6 bacteriophages from 54 isolates. In fact these phages once isolated from sewage/rivers were passaged four times on the Aeromonas P1A-XB1-6 propagation host. This passage of the bacteriophages through the propagation host helps to explain why later the 6 phages mainly infect Aeromonas sp. In L147-152 it is not surprising that the 6 phages generated more plaques in Aeromonas isolates, these phages were selected on different bacterial hosts but they were passaged x4 in Aeromonas P1A-XB1-6. Your experimental protocol biased towards this genus.

L162 Figure 2A and 2B need better explanation in the Results section. What is meant by "medium and low efficiency". All the EOPs are based on the standard strain? This is the same strain that all the phages were passaged through during purification. This strain becomes the experimental bias in the study.

L135-136 Plaque morphologies are poor indicators of anything. This data should not be used.

L149 no need to define PFU

L276 "cross-adaptation effect" ...this has more common terminology in phage biology. If using your own terms or from literature please define or reference.

Figure 6 - Figure should be deleted, plaque morphologies are poor indicators of host specificity. Yes they look different on hosts but they remain poor indicators of host range.

L355 "...we used taxonomically diverse bacteria native to this sample as potential hosts" Yes you did, but the error here is that all 6 phages were passaged through Aeromonas P1A-XB1-6 not their original isolation hosts.

L555 How many phage genomes were fully sequenced?

Figure 3 Virion morphology - only one electron micrograph is needed since both are the same.

Reviewer #2 (Comments for the Author):

Using a novel and meticulous approach, the authors isolated a group of closely related bacteriophages exhibiting the novel phenotype of extremely broad host ranges which they describe evocatively as "promiscuous." Their work was rigorous and exceptional care was taken to purify virus isolates and conduct experiments to avoid artifacts. Some of their findings are astonishing and the work has potentially significant implications for biotechnology and microbial ecology.

General comments and questions for the authors to consider.

1. I urge the authors to deposit these novel Atoyac phages and their bacterial hosts in a culture collection.
2. Did you record temperature, pH, salinity, or other parameters at the sampling sites?
3. The methods employed for bacterial host isolation seem likely to yield species that are fast-growing and adapted to high nutrient levels. This is not a criticism, but perhaps you would wish to specifically discuss how generalizable your results will be to other ecosystems. Are "contaminated" sites productive in terms of isolating promiscuous phage because the nutrient levels are elevated in those locations?
4. The incubation temperature for bacterial isolations was 30° C, was that true for all phage experiments as well? EOP is a convolution of several functions including the density of receptors, physiological status of the host and activity of restriction/CRISPR systems all of which are influenced by physicochemical conditions. How do the experimental conditions of temperature and nutrient levels you employed in your studies compare to those experienced by host cells in situ?
5. One thing to keep in mind is that both your viruses and host bacterial strains were fresh isolates from natural environments adapting to laboratory culture. Could that process explain why one of your host-phage pairs suddenly has an EOP that fell below the limit of detection?
6. Why were Atoyac 1 and Atoyac 23 phage chosen for evolution studies? If it is on the basis of subgrouping wouldn't Atoyac 13 or 15 have been a better choice based on sequence divergence?
7. The weak relationship of of Atoyac phage to other viruses/sequences in databases is interesting. Being able to deduce some putative ORF functions suggests relationships to other viruses or segments of them, albeit distant, will be discoverable.
8. The Atoyac phage are remarkably promiscuous with regard to host range. Compared to other systems in which phage infecting heterologous strains of the same bacterial species exhibit EOP declines of 10,000-fold or greater due to restriction endonucleases, the Atoyac viruses seem sometimes almost refractory to antagonism by the combined activities of restriction-modification systems, CRISPR or other host defenses. A brief review of the literature regarding the impact of restriction-modifications systems on EOP may help you set the context for what you have observed. If I am interpreting your results correctly, this is a noteworthy observation with potentially far-reaching ecological implications and direct biotechnological utility, something warranting detailed discussion.

Specific comments

1. Line 336 - the term "big majority" is certainly in the vernacular, but since it is difficult to define this notion, using the modifier big is redundant.
2. Line 479-80 - need definitions of abbreviations.
3. Line 481 - "somewhat randomly" is a paragon of methodological imprecision. The process was

either a random one or not. If not, inform readers what procedure you employed.

4. Line 487 and continuing - It appears you isolated bacteria using several media, but the description of phage assays is confusing. The first part of the description looks like a standard top agar protocol, but then you go on to mention LB, BHI, etc., so it is not clear what was done in terms of media usage. This is important because part of your success in recognizing broad host range phage hinged on your plaque assay protocols. Also, the abbreviations used for media are not defined or referenced. It is important to provide a precise account of your growth conditions, including incubation temperatures as these parameters could significantly impact the results or impair replication of your studies by others.

5. Line 535 - Spot titers were performed with 5-fold replication. What criterion did you use to decide which spots to enumerate? For example, did you count spots with 10-20 plaques? Lower numbers will increase the error term.

6. Table S2 - column header has "plate forming units" and I think you mean plaque forming units.

7. Figure 5 - The symbol colors make this data tricky to follow. Showing the number of times a batch of evolved virus exhibits a lower EOP compared to the reference host does not give the reader a sense of what this represents at the level of difference in plaque numbers. It is not clear what your cutoff points were - how different do two EOP values have to be to declare one lower than the other or equivalent to the reference value? A table or bar chart with averaged numeric EOP data might be easier for the reader to follow. In addition, the figure depicts lines as if the changes with passage vary as a continuous function. That may be, but you have only assessed three discrete time points in your studies.

Response to referees

Re: Dynamics of infection in a novel group of promiscuous phages and hosts of multiple bacterial genera retrieved from river communities

Reviewer #1

GENERAL COMMENTS

1) The title and abstract fail to describe the major discoveries of the study. Both are too broad and do not reflect the obtained results.

As per the reviewer's suggestion, we have now modified both the title and abstract of our manuscript, highlighting specific findings of our study, e.g. the isolation of a group of podophages with a remarkably broad host range within the Gammaproteobacteria, and their dynamic adaptation to hosts of different genera retrieved from river communities.

2) Authors are encouraged to use terminology that is accepted by majority of bacteriophage researchers. I highly encourage the authors to consult all of Steve Abedon's publications on bacteriophages, paying particular attention to his work on bacteriophage host range and the limitations of plaque assays in determining phage host range. Please do not over use the term "promiscuous" when in fact you are refereeing to broad host range bacteriophages.

We thank the reviewer for the valuable suggestions. We are familiar with the work of Steven Abedon and have tried to adhere to accepted phage research terminology whenever possible. As the reviewer points out, we are aware of the limitations of plaque assays to evidence phage infections, as this technique applies only to bacterial species that can grow as a confluent lawn [Kropinski et al. 2009, https://doi.org/10.1007/978-1-60327-164-6_7]. Additionally, these assays may not lead to the generation of lytic plaques with certain phages or under standard growth conditions (e.g. chronic infections or cases with reduced infection vigor), which would represent false-negative results in terms of identifying phage infection [Hyman and Abedon 2010, [https://doi.org/10.1016/s0065-2164\(10\)70007-1](https://doi.org/10.1016/s0065-2164(10)70007-1)]. Still, we decided to report the host range of our phages in the context of plaque assays because they provide concrete evidence for productive infections, despite the risk of underestimating this trait in some hosts. In contrast, the use of other techniques such as spotting can initially demonstrate bacterial killing but do not necessarily reflect phage propagation, thus leading to false positives and overestimation of host range [Mirzaei and Nilsson 2015, DOI: <https://doi.org/10.1371/journal.pone.0118557>]. Given the importance of convincingly demonstrating that Atoyac phages can productively infect hosts from distinct taxonomic ranks, we considered that proving plaque production was not only informative but necessary. Yet, we have now acknowledged in the introduction that plaque assays are susceptible to false positives (Lines 71-76). The term "broad host range" has been used to refer to phages infecting hosts of different species. However, it is also indistinctly used to describe phages with infection capabilities spanning multiple strains of the same species [Mirzaei and Nilsson 2015, DOI: <https://doi.org/10.1371/journal.pone.0118557>; Hyman 2019,

<https://doi.org/10.3390/ph12010035>]. To prevent confusion, we prefer the use of the term “promiscuous” to describe phages infecting bacteria from multiple taxonomic ranks. In biology, the term “promiscuous” has been applied to describe factors with a variety of functions, targets or hosts [e.g. Zhang et al. 2011, <https://doi.org/10.1371/journal.pbio.1000592>]. We choose this term because it has been applied to other types of mobile genetic elements, plasmids, where it refers to molecules that can be transferred and replicated in a broad range of host species [Krishnapillai 1988, [https://doi.org/10.1016/0378-1097\(88\)90003-1](https://doi.org/10.1016/0378-1097(88)90003-1); Iyer 1990, [https://doi.org/10.1016/0734-9750\(90\)90646-S](https://doi.org/10.1016/0734-9750(90)90646-S)]. Although to a lesser extent, “promiscuous” has also been used in the context of phages to describe broad specificity of infection or interaction amongst diverse host species [Ma et al. 2018, <https://doi.org/10.1186/s40168-018-0410-y>; Zborowsky and Lindell 2019, <https://doi.org/10.1073/pnas.1906897116>; Chatterjee et al. 2019, <https://doi.org/10.1128/IAI.00085-19>]. In lines 247-251 of the manuscript we briefly discuss why we favor the use of the term “promiscuous”, yet, we have limited its use to refer to the phages reported in our study only.

3) The manuscript contains non-standard terminology; terminology should be explained to the reader or referenced.

It was not clear to us what terminology was considered as non-standard by the reviewer, but we have made our best effort to explain or reference the terms that we used in our manuscript (see below in our response to the other points). We addressed the use of the term “promiscuous” in the previous point. The concept of cross-infection in the context of phages has been used to describe a method that evaluates the potential of phages to infect hosts of diverse environmental or taxonomic origins (different species or genera) [Weitz et al. 2013, <http://dx.doi.org/10.1016/j.tim.2012.11.003>; Chen et al. 2018, <https://doi.org/10.3389/fmicb.2018.01476>; Flores et al. 2011, <https://doi.org/10.1073/pnas.1101595108>]. In our study, we refer to cross-infection assays in the latter sense, i.e. the test aims to evaluate whether a phage infecting a host from a particular taxon can infect bacteria of different species or genera. We have now extended on the purpose of the cross-infection assays and how they were performed in lines 445-453 (Materials and Methods), and the legend of Supplemental Figure 1 (L803-808).

4) If I understand this correctly, out of all the bacteriophages isolated (54?) only 6 were sent for full genomic sequencing? If so, are the genomes deposited in GenBank? Materials section does not provide this information. Biological replicates should be mentioned in the M&M, there was a brief mention in a figure description.

We isolated six phages, which can infect 54 different bacterial isolates. We thank the reviewer for pointing this out, as we now realize this can be a source of misunderstandings; we clarified this in lines L112-114 of the revised version of our manuscript.

The complete genomes of the six isolated phages were generated in this study, and their sequences were already deposited in GenBank (the accession numbers can be found in the section “Sequence data availability”, L566-569). We have requested the

earlier release of the sequences, which has already been granted, and now they can be accessed.

Regarding the use of biological replicates, in the “Materials and Methods” section we indicate the number of replicates used to determine the EOP (Line 496), and during the evolution experiments (Line 564). Also, we have now added the number of replicates used in the coliform count assay (Line 419).

5) Bacterial genes *rpoD* and *gyrB* were used to bring the bacterial isolates down to genus level. In my experience and based on literature, 2 genes are not enough.

The use of phylogenetic markers for bacterial taxonomic identification to the level of genus and species is a conventional method. The 16S rRNA gene is the most common target for this purpose, and it is accepted by the Clinical and Laboratory Standards Institute (CLSI guideline MM18-A2, <https://clsi.org/standards/products/molecular-diagnostics/documents/mm18/>). However, it has been documented that due to the low sequence variability of this target gene, 16S rRNA can sometimes fail to discriminate between closely related species (17-35% error rate) or genera (up to 10% error rate) belonging to complex groups [Church et al. 2020, <https://doi.org/10.1128/CMR.00053-19>]. In contrast, the housekeeping genes *rpoD* and *gyrB* have a higher evolutionary rate and thus have shown better results on resolving complex groups such as *Pseudomonas*, Enterobacteria, and *Aeromonas* to the intrageneric level [Yamamoto et al. 2000, <https://doi.org/10.1099/00221287-146-10-2385>; Fukushima et al. 2002, <https://doi.org/10.1128/JCM.40.8.2779-2785.2002>; Navarro and Martinez-Murcia, <https://doi.org/10.1111/jam.13887>]. Multi-locus sequence analysis such as MLSA or MLST are recommended to discriminate with a high resolution between closely related species or to identify new species (by reconstructing a species phylogeny), which allows distinction even to the level of strains or epidemic clones [Gevers et al. 2005, <https://doi.org/10.1038/nrmicro1236>]. In our study, we used the sequences of the markers *rpoD* and *gyrB* to assess the phylogenetic relationships between the phage-susceptible bacterial isolates to determine the breadth of the host range of our phages. Despite the fact that these markers show considerably better discriminatory resolution than the 16S rRNA gene (Mulet et al. 2010, <https://doi.org/10.1111/j.1462-2920.2010.02181.x>), and have been used to resolve several intra-genus phylogenies, we limit our taxonomic assignments to the level of genus. The phylogenies presented in our work clustered all the phage-susceptible bacterial strains recovered from the environment with previously reported reference strains of different genera and species, robustly supporting their taxonomic assignment at the genus level (Figure 1). We also have 16S rRNA sequence data that support the taxonomic assignment inferred from the *rpoD* and *gyrB* sequence alignments; however, we do not report this information because it is not available for all the strains.

6) The Materials section is missing important information. While it should be succinct, nonetheless, the reader should be able to repeat and understand what was done and how it was done.

L489 What do all the media short terms refer to? What percentages of the media were used? Who is the manufacturer? L491 "Millipore membrane"????L495 is this from

some paper? Reference? L496 "cyclic and simultaneous cross-infection" definition?reference? L531 "Cross infection assay, see above"? L524 "Pelleting at 8,000 G" Meaning?

Thanks for bringing this to our attention. The "Materials and Methods" section has been greatly improved in this version of our manuscript after including information suggested by the reviewers. Relevant to this point, we have now included the full name of the media used. Difco is the manufacturer for all the media, which were prepared following the manufacturer's instructions as is now indicated in the "Materials and Methods" section citing the corresponding reference (Lines 422-427). The membranes that we used to filter the water samples were MF-Millipore™ Membrane Filters, 0.22 µm pore size; which is now indicated in line 439. The concentrations of the dyes used in our experiments follow those reported for Difco's standard media such as mFC and TSI [Difco & BBL Manual: Manual of Microbiological Culture Media. 2009. Becton Dickinson and Company]. We have now indicated this in lines 442-444.

Thank you for pointing out the confusion regarding the concepts "Cyclic and simultaneous cross-infection". As mentioned in point 3 of this document, we refer to cross-infection assay as a method to evaluate the potential of a phage to infect diverse hosts. By "Cyclic and simultaneous" we meant to highlight that the cross-infection assays were performed in two ways; selected individual phage plaques were transferred to bacterial lawns of different strains both 1) testing multiple bacterial isolates in parallel (simultaneously), and 2) picking a plaque produced in a single tested bacterial isolate and transferring it to a new strain (sequentially). We chose to test both approaches to generate stronger evidence for the phages' promiscuity and to assess whether this trait is lost after infecting a certain host at least once, which we did not observe. To avoid confusion, we have removed "Cyclic and simultaneous cross-infection" (both from the "Material and Methods" and the Supplemental Figure 1) and instead described the two different ways in which the assay was performed (Lines 445-451). In line 489 (before L531) we refer the reader to the Material and Methods, "Phage isolation" subsection where we briefly describe the cross-infection assays.

The relative centrifugal force (RCF) is a function of the rotor's radius and the revolutions per minute. Therefore, it is recommended to report the rate of centrifugation in terms of the g force (G or RCF) rather than revolutions per minute (RPM), which only express the speed of rotation and are therefore less precise. We have now replaced "pelleted" for "centrifuged" and changed the G unit to RCF because this term is more widely used (Line 482).

7) Samples were collected from sewage/river samples (L469-476). River water samples with high coliform counts used. Would not sewage have all the bacteriophages you need? I was not clear as to why the river waters were collected. In the rivers were you looking for human origin coliforms? Was this study

Our study aimed to identify phages capable of infecting bacteria from different taxonomic ranks. At the beginning of the study, we did not know where we could find this type of bacteriophages; hence, we decided to investigate distinct types of water samples: river and sewage. The purpose of counting thermotolerant coliforms in the

samples was to assess the impact of anthropogenic contamination on the sampled sites, as some of the rivers are located near to cities or are already being used for sewage discharge (See Supplemental Table 1). We hypothesised that polluted sites would feature higher bacterial densities (because of the high concentrations of nutrients introduced), and would therefore be richer sources for the isolation of phages. All the samples were collected at the same period of time and were then processed in parallel to build the collection of bacterial isolates that could be used as potential hosts for phage isolation. It was after the isolation of the Atoyac phages when we found that all of them were retrieved from contaminated water samples (Supplemental Table 1).

8) This manuscript has too many supplements. Supplements should be used for complex/detailed data that confirms main results - otherwise it should be simply 'data not shown'.

Following the reviewer's suggestion, we have made our best effort to reduce the items in the Supplemental Material. We have removed the Supplemental Figures 3, 4 and 5. Relevant information from the figures has now been integrated into either a main Figure or other Supplemental material, as suggested by the editor. For example, coordinates corresponding to the geographic location of the sampling sites where we identified Atoyac phages, previously indicated in Supplemental Figure 3, have been added to Supplemental Table 1. Likewise, micrographs of the Atoyac phages 10, 13, 14 and 23, before shown in Supplemental Figure 4, are now part of of the new Supplemental Figure 5 (previously numbered 8). The phylogeny of the Atoyac phages and their closest homologues, previously corresponding to Supplemental Figure 5, has now been incorporated into Figure 4. Hence, the number of Supplemental Figures was reduced from 8 to 5 in the new version of the manuscript. These remaining figures comply with the requirement of presenting detailed data supporting our findings or helping the reader to understand relevant experimental strategies.

9) In the Results sections authors should only describe the results that were described in the figures/data tables etc. The focus should be on the material in the main manuscript, not the supplements. L115 "phages isolated from river and sewage. (Supplement Fig. 3)" Is the reader really supposed to go to this? What is missing here in the main body of the results, is the number of phages isolated from river and numbers isolated from river which were further characterized in this work. L189-191 this should be in the discussion.

The purpose of the Supplemental Figure 3, referred in Lines 115-116, was to provide the reader with details about the geographical location of the sampled sites that were the source of Atoyac phages. Following the suggestion by the reviewer and the editor, we have removed the Supplemental Figure 3 and integrated the corresponding data in the Supplemental Table 1, which now includes the detailed information about the sampled sites: geographical location, type, the levels of coliforms recorded, and whether they were the source of Atoyac phages reported in our study. We have also indicated in the results which Atoyac phages were identified

in the different geographical regions and which were isolated from sewage or contaminated rivers (Lines 117-123).

Only the phages with the broadest host range against the panel of bacterial isolates tested in our screening were selected for further purification and characterization in this study, i.e. the six Atoyac phages. It is difficult to establish how many phages were present in the samples as we did not keep track of those infecting a single or few bacterial isolates. In Lines 112-114 (Results) and 451-453 (Materials and Methods), we have now clarified that only the six Atoyac phages were selected for further purification and characterization in this study.

As requested by the reviewer, the sentence in lines 187-191 has now been moved to the discussion (Lines 296-299).

SPECIFIC COMMENTS

L26 "bacteria isolated from the same samples", not clear what is meant.

Here what we meant is that bacteria used as hosts to identify and isolate the Atoyac phages were retrieved from the collection of water samples used to isolate phages. As mentioned in our response to general comment 1, we have now restructured the abstract, including a slight modification to this part (Line 26-28).

L32-33 and L94 "plating efficiency" or efficiency of plating (EOP)?

Thank you for bringing this to our attention. The two forms have been used interchangeably by Elizabeth Kutter [https://doi.org/10.1007/978-1-60327-164-6_14]; yet, to prevent confusion, we decided to homogenize to "efficiency of plating" or EOP.

L60-63 This is not a correct statement. Literature contains a significant number of publications that describe the ability of the Enterobacteriaceae phages to infect different genera.

The reviewer is right regarding the occurrence of Enterobacteriaceae phages infecting different genera. As several genera within this group are closely related phylogenetically (e.g. *Escherichia*, *Shigella*, *Salmonella*, *Citrobacter*), the taxonomic ranks do not seem to represent an important biological barrier for some phages (see Ackermann and Dubow (1987): "there is no such thing as a 'coliphage'; chances are 99% that it lyses *Klebsiella* and *Shigella* and somewhat less that it lyses *Salmonella* and *Proteus*"). Nevertheless, we do not consider the referred statement incorrect. Firstly, we do acknowledge previous reports on phages infecting different genera, both in the referred sentence: "Although phages capable of infecting bacteria beyond the taxonomic rank of species or genus have been previously described", and further below in the introduction (Lines 59-62) and the discussion (Lines 254-257). The second part of the sentence refers to the frequency of isolation of the so-called broad-host-range phages relative to narrow-host-range phages: "these are considered rare when compared to the far larger number of phages reported as specific to a single bacterial species". This is supported by published work from authors cited at the end of the sentence:

* Ackermann and Dubow (1987): "The record of phages crossing generic boundaries is small when compared to the very considerable number of genus-specific phages"

* Matsuzaki et al. (2005): “Phages are able to adsorb to specific bacterial species or to specific strains; phages capable of infecting across bacterial species or genera (so-called polyvalent phages) are few in number”

* Hyman and Abedon (2010): Here the authors quote the statement from Ackermann and Dubow and further explore “claims of phage host ranges spanning multiple genera”, finding limitations in the supporting evidence of several reports.

Likewise, literature cited in other parts of our manuscript support this notion on the frequency of isolation between the two phage types:

* Koskella and Meaden (2013): “There is clear evidence that not all bacteria are infected by all phages, and indeed that most phages can only infect a subset of bacterial species”

* De Jonge et al. (2018): “multiple phages with a broad host range, and their molecular mechanisms, have been identified, and ecological as well as metagenomics studies indicate that the broad host range may be more prevalent than previously thought. Yet, a narrow host range is prevalent among isolated and well-studied phages”

Still, we have rephrased the sentence slightly to make the message clearer.

L104 "infectivity assays" are actually spot tests.

We acknowledge that “infectivity assays” was ambiguous. Hence, we have removed it and rephrased the sentence to emphasize that serially diluted aliquots from the samples were “spotted” on lawns of the potential bacterial hosts, thus allowing the identification of phage plaques (Lines 106-108).

L105-106 What does this mean.

The water samples from the twelve sampled sites were first used to assemble a collection of bacterial strains (described in the “Materials and Methods” section) that could be used as potential hosts for phage isolation. In most cases, we detected clearing zones and phage plaques when we spotted aliquots of the water samples onto lawns of bacterial isolates from this collection. As mentioned in the previous point, we rewrote this sentence to clarify its message (Lines 106-108).

L107-109 Meaning of this sentence is not clear. This study simply relied on spot tests and toothpick streaks.

Indeed, this part of our study relied on spotting serial dilutions and toothpick streaks to identify and confirm plaque production as a proxy for productive infection, which we consider the result of a phage-bacteria interaction. In this sentence, we aimed to highlight that in some cases the use of traditional media (e.g. LB or TΦ) did not allow the identification of plaques, and that employing other media and/or dyes (as indicated in the Materials and Methods, “Phage isolation” subsection) led to plaques visualization. We have edited the sentence to make it easier to understand (Lines 112-114).

L109 What is "cross-infection screening"?

It refers to the implementation of a series of cross-infection assays (i.e. toothpick streaks from one phage plaque onto multiple bacterial lawns) to examine the phages' ability to infect multiple bacteria and identify those with a broad host range. As we mentioned in our response to the general comments 3 and 6, cross-infection assays are now described in the "Materials and Methods" section and Supplemental Figure 1, to which the reader is referred in this sentence (Lines 445-453).

L111 The study actually focuses on 6 bacteriophages from 54 isolates. In fact these phages once isolated from sewage/rivers were passaged four times on the Aeromonas PIA-XB1-6 propagation host. This passage of the bacteriophages through the propagation host helps to explain why later the 6 phages mainly infect Aeromonas sp. In L147-152 it is not surprising that the 6 phages generated more plaques in Aeromonas isolates, these phages were selected on different bacterial hosts but they were passaged x4 in Aeromonas P1A-XB1-6. Your experimental protocol biased towards this genus.

From the panel of bacterial isolates retrieved from the water samples, *Aeromonas sp.* PIA_XB1_6 was selected as the propagation host because, during the first screening to identify phage-susceptible strains, we observed higher titers, larger plaques, and shorter time periods of plaque formation in this bacterial isolate compared to other strains. Other highly susceptible strains were also *Aeromonas sp.* We failed to include this information in the manuscript, but it has now been added to the "Material and Methods" section (Lines 453-458). Still, we agree with the reviewer and acknowledge that the higher EOP observed in *Aeromonas* could be explained by the process of plaque purification carried out in this host. Unfortunately, this is an artifact inherent to the protocols to obtain pure phage stocks. As highlighted by Hyman and Abedon (2010): "...plaque purification, minimally three rounds, is a crucial precaution before host-range determination...". Hence, we prioritised obtaining pure phage stocks over the potential for a domestication effect in the propagation host. We have added a sentence in the "Discussion" section (Lines 312-319) acknowledging this source of bias and its potential consequence.

L162 Figure 2A and 2B need better explanation in the Results section. What is meant by "medium and low efficiency" . All the EOPs are based on the standard strain? This is the same strain that all the phages were passaged through during purification. This strain becomes the experimental bias in the study.

The aim of adding "high, medium, and low efficiencies" labels in panel A of Figure 2 was to help the readers in identifying the isolates in which the Atoyac phages produced more plaques with an easy-to-interpret categorization. These categories were assigned based on the tree (hierarchical clustering) inferred from the EOP values. However, we acknowledge such categorization was somewhat arbitrary and unnecessary since an EOP scale is provided at the bottom of the figure. Thus, we have removed the labels in the new version of the figure. We calculated the EOPs using as reference *Aeromonas* PIA_XB1_6, the propagation host. As we explained in the previous point, this environmental strain retrieved from the water samples was chosen as the propagation host because we observed it was highly susceptible to phage infection during our initial examination to identify phage-

susceptible strains, i.e. prior to the purification of the Atoyac phages. We limited the propagation in PIA_XB1_6 to the minimum number of steps to reduce the risk of a domestication effect (Lines 457-458), nevertheless, we have acknowledged the potential impact of propagation in this strain on the observed EOPs (see above). Still, our determination of EOP in the panel of susceptible strains showed a range of efficiencies among *Aeromonas*: at least two other strains displayed an EOP higher than the propagation host, whereas several others featured efficiencies lower than PIA_XB1_6 and strains of genera different to *Aeromonas*. This observation suggests that purification in PIA_XB1_6 is not the only factor contributing to the EOP observed in a particular host or genus.

****L135-136 Plaque morphologies are poor indicators of anything. This data should not be used.**

Our main goal by presenting Supplemental Figure 2 was to provide concrete evidence supporting our observations on the Atoyac phages breadth of host range. As summarized by Hyman and Abedon (2010): “Unexpected observations demand rigorous scrutiny, and such scrutiny should be especially the case when demonstration of phage infection of particular hosts is the emphasis of a publication”...“Note also that some authors refer to the demonstration of bacteria killing, without associated phage production, as bacteriocin- or lysis from without-like activity”. Hence, the lack of evidence for productive infection has been a main point of criticism for several reports on broad-host-range phages, especially when the host range is remarkably broad, as it is the case of the Atoyac phages. We respectfully disagree with the reviewer's view on plaque morphologies being poor indicators of anything; interesting work on the theory of phage plaques by Abedon and Yin (2019; https://doi.org/10.1007/978-1-60327-164-6_17) highlights the relevance of plaque growth in the study of phages and thoroughly discusses factors affecting plaque formation and the relationship between plaque development and different stages of phage infection. Nonetheless, we acknowledge there is controversy regarding the comparison of plaque morphologies given the multiple factors that influence this trait. Thus, we have removed all sentences in the manuscript related to our observations on differences between plaque morphologies (Lines 136-139, 291-302, Lines 325-331; see below our response to the comment on Figure 6), but we decided to keep the Supplemental Figure 2 since its main purpose is to provide tangible evidence of the Atoyac phages ability to form plaques in host of different taxonomic origin as a proxy for productive infection.

L149 no need to define PFU

We have deleted this.

L276 "cross-adaption effect" ...this has more common terminology in phage biology. If using your own terms or from literature please define or reference.

We thank the reviewer for the comment. We were not able to find terminology describing the observed effect (propagation in a single host improving EOP in a taxonomically unrelated strain), that was common in phage research literature. In a

personal communication with an expert on bacteriophage evolution, Professor Michael Brockhurst from the University of Manchester, he suggested referring to this phenomenon as “evolution of expanded host range”, which we have done now (Lines 223-225).

Figure 6 - Figure should be deleted, plaque morphologies are poor indicators of host specificity. Yes they look different on hosts but they remain poor indicators of host range.

Our intention when preparing Figure 6 was to show that changes in EOP observed in our evolution experiments were, in some instances, accompanied by considerable fluctuation in plaque morphology. These observations are reproducible under our experimental conditions and we believed could be of interest to the reader. However, we appreciate the reviewer's suggestion and Figure 6 has now been deleted. Accordingly, this has led to removal of several lines of text in the Results and Discussion sections of our manuscript (Lines 291-302, Lines 325-331).

L355 "...we used taxonomically diverse bacteria native to this sample as potential hosts" Yes you did, but the error here is that all 6 phage were passaged through *Aeromonas* P1A-XB1-6 not their original isolation hosts.

We realise there might have been some confusion regarding the *Aeromonas* strain PIA_XB1_6 that we call “reference strain”. We call it this way because we used it as a reference to calculate the EOPs. This, however, is not a laboratory reference strain, it was retrieved from the water samples we isolated phages from. Hence, PIA_XB1_6 is part of the panel of environmental strains reported in this study as susceptible to infection by Atoyac phages. We have now clarified this in the manuscript (Lines 131-134).

Purification by at least three passages in a single strain is deemed as required prior to reporting a phage host range to avoid false positives arising from phage mixtures [Hyman and Abedon 2010, [https://doi.org/10.1016/s0065-2164\(10\)70007-1](https://doi.org/10.1016/s0065-2164(10)70007-1)]. It is difficult to establish what the original host of the Atoyac phages is, but from the panel of susceptible environmental strains, we chose PIA_XB1_6 as the propagation host because it produced plaques in short periods of time and high titers (See Materials and Methods).

In the referred paragraph by the reviewer (Line 355, now 261) we discuss that instead of using enrichment methods with laboratory reference strains to isolate broad-host-range phages, we identified them by using a diverse collection of environmental strains as potential hosts. This statement holds true irrespective of which environmental strain was chosen as propagation host, since the Atoyac phages are still able to infect taxonomically diverse bacteria from the sampled sites.

L555 How many phage genomes were fully sequenced?

Thank you for the observation. In this study, we whole-genome sequenced the 6 Atoyac phages. This information has now been added to the manuscript (Lines 512-517).

Figure 3 Virion morphology - only one electron micrograph is need since both are the same.

As per the reviewer's suggestion, only the micrograph of Atoyac phage 1 was kept as representative of the group. The micrographs of the other Atoyac phages were integrated into the Supplemental Figure 5.

Reviewer #2

Comments for the Author:

Using a novel and meticulous approach, the authors isolated a group of closely related bacteriophages exhibiting the novel phenotype of extremely broad host ranges which they describe evocatively as "promiscuous." Their work was rigorous and exceptional care was taken to purify virus isolates and conduct experiments to avoid artifacts. Some of their findings are astonishing and the work has potentially significant implications for biotechnology and microbial ecology.

General comments and questions for the authors to consider.

1. I urge the authors to deposit these novel Atoyac phages and their bacterial hosts in a culture collection.

We thank the reviewer for the interest in the phages and bacterial strains reported in our study. We are currently reviewing the process to submit them to the ATCC or NCTC. However, in the meantime, we are committed to share any biological material reported in this study upon request.

2. Did you record temperature, pH, salinity, or other parameters at the sampling sites?

In addition to estimating coliform levels, we recorded some physicochemical parameters in situ for most of the river samples, including temperature, pH, salinity, redox potential, dissolved oxygen, and total dissolved solids. This has now been indicated in the "Materials and Methods" section (L412-414) and the data have been added to the Supplemental Table 1.

3. The methods employed for bacterial host isolation seem likely to yield species that are fast-growing and adapted to high nutrient levels. This is not a criticism, but perhaps you would wish to specifically discuss how generalizable your results will be to other ecosystems. Are "contaminated" sites productive in terms of isolating promiscuous phage because the nutrient levels are elevated in those locations?

This is an interesting and important observation. Indeed, the strategy we used shows that promiscuous phages were retrieved from contaminated sampling sites. As mentioned by the reviewer, we believe our bacterial isolation strategy was particularly successful, and likely biased, towards recovering fast-growing bacteria adapted to exploit the abundant resources available in these ecosystems impacted by human discharges. We speculate that in these environments there is a greater microbial diversity in high-density populations, which would therefore favor the presence of phages capable of infecting different potential hosts. However, we do not rule out that promiscuous phages could also be recovered from cleaner environments, but this type of sample might require an additional enrichment step to yield phage titers at a level detectable by the current microbiology techniques.

We think our growth conditions could be applied to samples obtained from similar ecosystems. However, we believe that the crucial point in our strategy to isolate promiscuous phages does not lie in using the exact same growth conditions that we used but in the isolation of bacteria native to a given environment as potential hosts for the phages with which they cohabit. Therefore, growth conditions that mimic the natural environment of the samples must be explored for each ecosystem from which researchers wish to isolate this type of phages. For instance, isolation media (media composition, pH), growth conditions (temperature, oxygen concentrations) should be explored and standardized to obtain as many native bacterial isolates as possible in order to maximize the possibility to find promiscuous phages.

As requested by the reviewer, we have added some lines to the discussion on this topic (L267-275).

4. The incubation temperature for bacterial isolations was 30° C, was that true for all phage experiments as well? EOP is a convolution of several functions including the density of receptors, physiological status of the host and activity of restriction/CRISPR systems all of which are influenced by physicochemical conditions. How do the experimental conditions of temperature and nutrient levels you employed in your studies compare to those experienced by host cells in situ?

We thank the reviewer for the observation. Yes, the incubation temperature for all the experiments was 30°C (we have now clarified this in the "Phage Isolation" section, line 441). We found that the average temperature in the different sampled sites was around 25°C, being in some site as high as 31°C. Based on this and in the observation that bacterial isolates grew fast both in liquid cultures and in plates during plaque assays when incubated at 30°C, we decided to use this temperature for all assays. The pH was practically neutral in all the rivers, which was the same pH to which we adjusted all the media. The average observed values for dissolved oxygen (5.9 ppm and 81% saturation) suggested that, in general, most microorganisms could be grown under normal oxygen levels, and therefore we used this condition. Regarding the nutrient levels, we speculate that the concentration could be high in many of the rivers, since most of them had a high coliform count as a consequence of the constant discharge of human waste.

Therefore, we believe that the growth conditions we used in the laboratory to explore the interactions between the Atoyac phages and their possible hosts, although not identical, are comparable to the environmental ones of their isolation sites in terms of temperature, pH and nutrients concentration (but perhaps not composition).

5. One thing to keep in mind is that both your viruses and host bacterial strains were fresh isolates from natural environments adapting to laboratory culture. Could that process explain why one of your host-phage pairs suddenly has an EOP that fell below the limit of detection?

This is a good observation. In our evolution experiments, we de-coupled the evolution of the virus and the bacteria by separating them in every transfer and replacing the bacteria with the ancestor growing up from a frozen glycerol stock (Supplemental Figure 4). Thus, in our experiments only the virus was allowed to evolve. Likewise, the EOPs were determined using fresh cultures of the ancestral host grown from glycerol stocks. We have now clarified this in relevant sections in the revised version of the manuscript (Lines 215-216, 780-782, 558-560). We therefore hypothesize that reduction in EOP below the limit of detection observed in phage Atoyac 23 and the *Pseudomonas* host is the result of changes in the phage arising from its propagation in the *Yersinia* isolate. Although we agree that laboratory conditions may alter host-phage interactions, potentially explaining the strong EOP reduction, we only observed this phenomenon in the phage-host combination mentioned above. Under the same culture conditions, and using ancestor bacteria from the same overnight batch culture, evolved versions of phage Atoyac 23 maintained a detectable EOP in the *Pseudomonas* strain when propagated in this host or when the host was substituted cyclically (*Yersinia*, *Aeromonas*, *Pseudomonas*) (Figure 5, panels B1 and B3). Furthermore, ancestral stocks of Atoyac 23 consistently exhibited the original EOP on bacterial lawns prepared using the *Pseudomonas* culture batch in which the *Yersinia*-propagated stocks displayed EOP reduction (data not shown). We also know that the *Yersinia*-propagated Atoyac 23 stocks contain phage because they rendered significant EOP values in lawns of the *Yersinia* isolate (Figure 5, panel B2). For Atoyac 1, no reduction in EOP was observed in any of the hosts or propagation schemes under the same culture conditions (Figure 5, panels A1-A3).

6. Why were Atoyac 1 and Atoyac 23 phage chosen for evolution studies? If it is on the basis of subgrouping wouldn't Atoyac 13 or 15 have been a better choice based on sequence divergence?

The reviewer is correct, the best representatives for the evolution experiments in terms of sequence divergence would have been Atoyac phages 1 and 15, which share ~86% overall nucleotide identity. Atoyac 15, however, corresponds to an outlier, both in terms of genetic distance and host range (it is the Atoyac phage infecting the fewest number of Enterobacteriaceae strains), thus we were unsure whether observations in this phage would be representative of the group. From the practical point of view, we also considered that when sequencing the genomes of the evolved phages in a follow-up project, it would be more difficult to link genotype-phenotype changes between phages due to the greater divergence to Atoyac 15. We

then decided to pick the most divergent representatives from the other sub-groups; hence the selection of phages 1 and 23, sharing ~95% sequence identity. Moreover, our goal was to compare phages featuring distinct EOP profiles on the strains selected for the experiments to assess whether they could overcome low efficiencies of plating in different hosts. For the chosen *Pseudomonas* isolate, Atoyac 1 exhibits a slightly higher EOP than phage Atoyac 23, whereas the opposite is observed for the selected *Yersinia* strain (Figure 5, propagation step 0). We have now added a sentence to the "Materials and Methods" section (Lines 553-556) briefly describing why phages 1 and 23 were selected.

7. The weak relationship of Atoyac phage to other viruses/sequences in databases is interesting. Being able to deduce some putative ORF functions suggests relationships to other viruses or segments of them, albeit distant, will be discoverable.

This is correct. When searching for homologs to the Atoyac phages at genomic level, we could only detect significant nucleotide sequence similarity to two *Yersinia* and some *Aeromonas* phages (in our comparative maps we only use the closest representative from each genus). However, using as query the amino acid sequences of certain ORFs with annotated functions, such as those from the structural or replication modules, allows the detection of remote homologous ORFs from various phages, concurring with the mosaicism typically observed in phage genomes. Interestingly, several matches correspond to phages with hosts of genera identified as susceptible to Atoyac phages, e.g. *Pseudomonas*, *Serratia* or *Escherichia*. Matches to phages with hosts of different genera were also observed (e.g. *Pectobacterium*, *Cronobacter*, *Salmonella*); thus, it would be interesting to test bacteria from these genera as potential hosts for Atoyac phages in a subsequent study. It is worth noting, however, that all matches to ORFs from phages with hosts other than *Yersinia* and *Aeromonas*, feature very low levels of sequence similarity, indicating very distant relationships. We have added a couple of sentences on this topic to the discussion section in the revised version of our manuscript (Lines 367-371).

8. The Atoyac phage are remarkably promiscuous with regard to host range. Compared to other systems in which phage infecting heterologous strains of the same bacterial species exhibit EOP declines of 10,000-fold or greater due to restriction endonucleases, the Atoyac viruses seem sometimes almost refractory to antagonism by the combined activities of restriction-modification systems, CRISPR or other host defenses. A brief review of the literature regarding the impact of restriction-modifications systems on EOP may help you set the context for what you have observed. If I am interpreting your results correctly, this is a noteworthy observation with potentially far-reaching ecological implications and direct biotechnological utility, something warranting detailed discussion.

We thank the reviewer for the insightful and valuable observation. Bacteria often have intracellular defense systems that allow them to limit infection by phages and other mobile genetic elements. Among the most widespread phage defense system in bacterial genomes are the Restriction-Modification Systems, and the CRISPR-cas

systems, which have been shown to dramatically impact the EOP by several orders of magnitude (Elizabeth Kutter, 2009 [https://doi.org/10.1007/978-1-60327-164-6_14], Hyman and Abedon, 2010, [https://doi.org/10.1016/s0065-2164\(10\)70007-1](https://doi.org/10.1016/s0065-2164(10)70007-1)).

Interestingly, as the reviewer mentioned, the Atoyac phages did not show a considerable reduction in the EOPs in most of their hosts, even among the different genera (~85% of the isolates showed less than 100-fold EOP variations, Figure 2). A previous study by Jensen et al. (1998) [PMID: [9464396](https://pubmed.ncbi.nlm.nih.gov/9464396/)] showed that broad-host-range phages that do not display significant decreases in their EOPs in different hosts (such as *S. natans* and *P. aeruginosa*) are insensitive to type I and II restriction enzymes. Therefore, it is possible that one or multiple of the various putative ORFs of unknown function that were found during the genomic annotation and comparison in the Atoyac phages could encode mechanisms to counteract anti-phage defense systems (such as RMSs, CRISPR, or others). We believe that by studying this type of phage, interesting proteins with biotechnology applications can be found, and it remains of our interest to study more in detail what these ORFs of unknown function encode. An example of this is the mechanism used to counter-attack bacterial defense by the also broad-host-range podophage T7, which encodes the Orc protein that inhibits type I restriction enzymes (Molineux (2006). The T7 group. In: Calendar R (ed). *The Bacteriophages*. Oxford University Press: New York, NY, USA, pp 277–301, Häuser et al., 2012 [<https://doi.org/10.1016/B978-0-12-394438-2.00006-2>]).

We have added some lines to the discussion regarding this (L336-352).

Specific comments

1. Line 336 - the term "big majority" is certainly in the vernacular, but since it is difficult to define this notion, using the modifier big is redundant.

We thank the reviewer for the suggestion. The correction has been applied (Line 242).

2. Line 479-80 - need definitions of abbreviations.

We have now included the definitions that were missing (Lines 423-427).

3. Line 481 - "somewhat randomly" is a paragon of methodological imprecision. The process was either a random one or not. If not, inform readers what procedure you employed.

We thank the reviewer for the comment. To isolate diverse bacteria, we picked colonies representative of the wide variety of morphologies (including size and color), observed in the plates of the different media used. We have then removed "somewhat randomly" and adjusted the sentence accordingly (Lines 427-428).

4. Line 487 and continuing - It appears you isolated bacteria using several media, but the description of phage assays is confusing. The first part of the description looks

like a standard top agar protocol, but then you go on to mention LB, BHI, etc., so it is not clear what was done in terms of media usage. This is important because part of your success in recognizing broad host range phage hinged on your plaque assay protocols. Also, the abbreviations used for media are not defined or referenced. It is important to provide a precise account of your growth conditions, including incubation temperatures as these parameters could significantly impact the results or impair replication of your studies by others.

We thank the reviewer for the kind comments, we agree that detailing this information is relevant. The abbreviated media in this section were already defined in the previous section "Bacteria Isolation", and the appropriate reference was also added. The purpose of testing several media on the bacterial lawns was to find the best conditions for identifying phage infection. A more detailed description of the phage assays has been included in the "Phage Isolation" section.

5. Line 535 - Spot titers were performed with 5-fold replication. What criterion did you use to decide which spots to enumerate? For example, did you count spots with 10-20 plaques? Lower numbers will increase the error term.

We thank the reviewer for the comment. We agree, the criterion that we always use is to count plaques in the spot with at least 20 plaques.

6. Table S2 - column header has "plate forming units" and I think you mean plaque forming units.

We thank the reviewer for the correction. This has already been changed.

7. Figure 5 - The symbol colors make this data tricky to follow. Showing the number of times a batch of evolved virus exhibits a lower EOP compared to the reference host does not give the reader a sense of what this represents at the level of difference in plaque numbers. It is not clear what your cutoff points were - how different do two EOP values have to be to declare one lower than the other or equivalent to the reference value? A table or bar chart with averaged numeric EOP data might be easier for the reader to follow. In addition, the figure depicts lines as if the changes with passage vary as a continuous function. That may be, but you have only assessed three discreet time points in your studies.

We thank the reviewer for the valuable suggestions. We admit that the figure was slightly confusing and in order to make it clear, we have changed the colours of the different lineages to more contrasting shades but we kept the overall colour selection to be consistent with the ones used in the other figures. Also, we agree that lines to follow the trajectories of the passages can be confusing, and therefore they have been removed.

Since the approach with the experimental evolution was to evaluate whether the efficiency of the Atoyac phages would maintain its original trend after being propagated in *Pseudomonas*, *Yersinia* or in multiple hosts, we thought that showing the variation in the values of EOP as "fold change" would make the comparison

easier to understand, however we have now realized that this was not the case. Therefore, in response to the reviewer's suggestion, these values have been replaced by those of EOP. Nonetheless, due to the complexity of the data, we believe that a bar chart is not the best option to convey a clear message, since many sets of bars would have to be displayed for each phage lineage and that could make the figure confusing. Additionally, we have added all the values of EOP estimated in the evolution experiments to Supplemental Table 5.

December 30, 2020

Dr. Daniel Cazares
Center for Genomic Sciences, UNAM
Genome Engineering
Cuernavaca, Morelos 62210
Mexico

Re: mSystems00773-20R1 (A novel group of promiscuous podophages infecting diverse Gammaproteobacteria from river communities exhibits dynamic inter-genera host adaptation)

Dear Dr. Daniel Cazares:

Below you will find a few minor comments of the two original reviewers who have thoroughly evaluated the manuscript. Once these comments are addressed, I will editorially accept your paper. Warm wishes for the New Year.

To submit your modified manuscript, log onto the eJP submission site at <https://msystems.msubmit.net/cgi-bin/main.plex>. If you cannot remember your password, click the "Can't remember your password?" link and follow the instructions on the screen. Go to Author Tasks and click the appropriate manuscript title to begin the resubmission process. The information that you entered when you first submitted the paper will be displayed. Please update the information as necessary. Provide (1) point-by-point responses to the issues raised by the reviewers as file type "Response to Reviewers," not in your cover letter, and (2) a PDF file that indicates the changes from the original submission (by highlighting or underlining the changes) as file type "Marked Up Manuscript - For Review Only."

Due to the SARS-CoV-2 pandemic, our typical 60 day deadline for revisions will not be applied. I hope that you will be able to submit a revised manuscript soon, but want to reassure you that the journal will be flexible in terms of timing, particularly if experimental revisions are needed. When you are ready to resubmit, please know that our staff and Editors are working remotely and handling submissions without delay. If you do not wish to modify the manuscript and prefer to submit it to another journal, please notify me of your decision immediately so that the manuscript may be formally withdrawn from consideration by mSystems.

Sincerely,

Seth Bordenstein

Editor, mSystems

Journals Department
Reviewer comments:

Reviewer #1 (Comments for the Author):

Few minor points that will improve the manuscript:

1) Figure 3 - centre the phage in the electron micrograph so that the blank space to the right is deleted.

2) Literature Cited needs editing:

L608 volume? page?

L610 volume? page?

Italics in the manuscript should be used as in the original publication ex. L611-613 italics missing on genus/species. Check entire section for this issue.

Style of the cited publications is not consistent: a) Majority use short forms for journal titles yet some are written in long form. Please check for consistency, b) Some article titles use capital letters majority do not ex. capitals are in L614-615 and L653-655

Reviewer #2 (Comments for the Author):

I urge the authors to make every effort to deposit their virus/host isolates in a culture collection. I do not question their intentions to provide samples on request, but point out that it is easy to lose track of stocks with the passage of time. More work is necessary, but some of the phages exhibit host defense evasion capacities with potential for future biotechnology applications. Captured from natural environments, if lost, recovery of these strains may not be a simple matter. Depositing samples in a culture collection may be an important means to increase awareness of your work and facilitate additional studies.

Response to referees

Re: A novel group of promiscuous podophages infecting diverse Gammaproteobacteria from river communities exhibits dynamic inter-genera host adaptation

Reviewer #1 (Comments for the Author):

Few minor points that will improve the manuscript:

1) Figure 3 - centre the phage in the electron micrograph so that the blank space to the right is deleted.

As suggested by the Reviewer, the blank space to the right of Figure 3 was deleted so that the phage virions in the micrograph are visualized centered.

2) Literature Cited needs editing:

L608 volume? page?

L610 volume? page?

Italics in the manuscript should be used as in the original publication ex. L611-613 italics missing on genus/species. Check entire section for this issue.

Style of the cited publications is not consistent: a) Majority use short forms for journal titles yet some are written in long form. Please check for consistency, b) Some article titles use capital letters majority do not ex. capitals are in L614-615 and L653-655

Thanks for bringing this to our attention. We added the missing information to the references indicated by Reviewer. We also checked the entire section to make sure that all References are complete, that genera, species and genes are appropriately displayed in italics, and that the article titles are consistent in the use of the short format and do not use capitals.

Reviewer #2 (Comments for the Author):

I urge the authors to make every effort to deposit their virus/host isolates in a culture collection. I do not question their intentions to provide samples on request, but point out that it is easy to lose track of stocks with the passage of time. More work is necessary, but some of the phages exhibit host defense evasion capacities with potential for future biotechnology applications. Captured from natural environments, if lost, recovery of these strains may not be a simple matter. Depositing samples in a culture collection may be an important means to increase awareness of your work and facilitate additional studies.

We agree with the Reviewer that the Atoyac phages and their hosts should be deposited in a culture collection to safeguard them and facilitate additional studies and appreciate the encouragement to do so. Since the first revision, we have tried

to contact the ATCC to deposit our material, unsuccessfully. This time, besides filling-up a deposit request from the ATCC, we have contacted the NCTC and the Félix D'Hérelle Reference Center for Bacterial Viruses with an inquiry to deposit our samples. We did not receive an answer from the NCTC so far either. However, we have now received a response from the Félix D'Hérelle Reference Center informing us of their interest to accept our biological material into their collection. The request we made contemplates the deposit of the six Atoyac phages and three bacterial strains different genera corresponding to the main host used in our study: *Aeromonas* sp. PIA_XB1_6, *Yersinia* sp. PAF_XS2_2 and, *Pseudomonas* sp. mFC_SR91. In the following days, we will work in completing the deposit process. We commit to adding the repository information to the manuscript as soon as this is available.

January 13, 2021

Dr. Daniel Cazares
Center for Genomic Sciences, UNAM
Genome Engineering
Cuernavaca, Morelos 62210
Mexico

Re: mSystems00773-20R2 (A novel group of promiscuous podophages infecting diverse Gammaproteobacteria from river communities exhibits dynamic inter-genera host adaptation)

Dear Dr. Daniel Cazares:

Your manuscript has been editorially accepted, and I am forwarding it to the ASM Journals Department for publication. For your reference, ASM Journals' address is given below. Before it can be scheduled for publication, your manuscript will be checked by the mSystems senior production editor, Ellie Ghatineh, to make sure that all elements meet the technical requirements for publication. She will contact you if anything needs to be revised before copyediting and production can begin. Otherwise, you will be notified when your proofs are ready to be viewed.

Sincerely,

Seth Bordenstein
Editor, mSystems

Journals Department
Table S2: Accept
Figure S3: Accept
Table S1: Accept
Figure S2: Accept
Table S3: Accept
Figure S1: Accept
Table S4: Accept
Figure S5: Accept
Figure S4: Accept
Table S5: Accept